# Extrinsic and intrinsic signals converge on the Runx1/CBFβ transcription factor for nonpeptidergic nociceptor maturation

**Siyi Huang[1,2], Kevin J O'Donovan[3†], Eric E Turner[4], Jian Zhong[3], David D Ginty[1,2]***

[1]Department of Neurobiology, Howard Hughes Medical Institute, Harvard Medical School, Boston, United States; [2]Department of Neuroscience, Howard Hughes Medical Institute, Johns Hopkins School of Medicine, Baltimore, United States; [3]Burke Medical Research Institute, Weill Medical College of Cornell University, White Plains, United States; [4]Seattle Children's Hospital, Seattle Children's Research Institute, Seattle, United States

**Abstract** The generation of diverse neuronal subtypes involves specification of neural progenitors and, subsequently, postmitotic neuronal differentiation, a relatively poorly understood process. Here, we describe a mechanism whereby the neurotrophic factor NGF and the transcription factor Runx1 coordinate postmitotic differentiation of nonpeptidergic nociceptors, a major nociceptor subtype. We show that the integrity of a Runx1/CBFβ holocomplex is crucial for NGF-dependent nonpeptidergic nociceptor maturation. NGF signals through the ERK/MAPK pathway to promote expression of *Cbfb* but not *Runx1* prior to maturation of nonpeptidergic nociceptors. In contrast, transcriptional initiation of *Runx1* in nonpeptidergic nociceptor precursors is dependent on the homeodomain transcription factor Islet1, which is largely dispensable for *Cbfb* expression. Thus, an NGF/TrkA-MAPK-CBFβ pathway converges with Islet1-Runx1 signaling to promote Runx1/CBFβ holocomplex formation and nonpeptidergic nociceptor maturation. Convergence of extrinsic and intrinsic signals to control heterodimeric transcription factor complex formation provides a robust mechanism for postmitotic neuronal subtype specification.

*For correspondence:
david_ginty@hms.harvard.edu

Present address: †Department of Chemistry and Life Sciences, United States Military Academy, West Point, United States

## Introduction

Neuronal cell fate specification is a multistep process that can be broadly divided into two stages, early specification of neural progenitor cell identity and postmitotic differentiation of neuronal subtypes. A recurring theme in the early phase of neural progenitor specification of the vertebrate nervous system is that the interplay between intrinsic determinants and extrinsic signals governs neural cell fate decisions (*Fishell and Heintz, 2013*). Indeed, cell fate choice of progenitor cells is determined by a combination of intrinsic genetic programs, which define the differentiation potential or the competence state of progenitors, and extrinsic cues, which influence the relative proportions of each cell type generated within the confines of a given competent state (*Livesey and Cepko, 2001*). Our understanding of the mechanisms of postmitotic differentiation of neuronal subtypes, on the other hand, is rather limited, and thus the relative contributions of extrinsic and intrinsic signals, and their mechanisms of interplay during late phases of cell fate determination, remain poorly understood.

Primary sensory neurons of the dorsal root ganglion (DRG) are pseudo-unipolar neurons that extend one axonal branch to the spinal cord and the other to peripheral targets such as the skin. The existence of a large number of functionally specialized DRG sensory neuron subtypes, each endowed with unique morphological and physiological properties enables the somatosensory system to encode diverse sensory information, including nociception, temperature, light touch, limb

**eLife digest** Animals detect and respond to their environment using their sensory nervous system, which forms through a complex, multi-step process. A precursor nerve cell's fate is set early in its development, and determines the different nerve types it can become. As development progresses, sensory nerve cells develop further into distinct subtypes that perform particular tasks, such as responding to touch or pain.

Nociceptors are the specialised sensory nerves that respond to potentially harmful stimuli. They form two distinct subtypes: peptidergic nerves detect potentially dangerous temperatures, whereas non-peptidergic nerves detect potentially dangerous mechanical sensations. Both subtypes originate from the same precursor nerve cell and both initially depend on an external molecule called NGF for their development and survival. During their development, non-peptidergic neurons stop responding to NGF and start producing a protein called Runx1, considered to be the 'master regulator' of non-peptidergic nerve cell development. Runx1 works by forming a complex with another protein called CBFbβ, and this complex activates a program of gene expression that is specific to non-peptidergic nerves. However it was unclear how an external signal, like NGF, can coordinate with or influence a nerve cell's internal genetic program during the nerve's development. It was also not known whether NGF and Runx1 interact with each other.

By studying non-peptidergic nerve cell development in mice that lack NGF, Runx1 and other associated proteins, Huang et al. have now established the sequence of events that regulate the development of this nerve cell subtype. Two signalling pathways converge to switch on non-peptidergic nerve cell development. An NGF-driven signalling pathway activates the production of CBFβ, while another protein binds to the Runx1 gene to switch it on. This leads to the production of the Runx1 and CBFβ proteins that complex together to activate the non-peptidergic neuronal genetic program.

These findings demonstrate how two different mechanisms converge to produce the component parts of a complex, which then activates a genetic program that drives the development of a particular neuronal subtype. Whether this mechanism is involved in determining the fate of other cell types remains a question for future work.

movement, and body position. DRG neurons can be classified into three major classes: nociceptors, which preferentially respond to noxious stimuli, pruritogenic stimuli, and temperature; mechanoreceptors, which respond to innocuous tactile sensations; and proprioceptors, which detect position and movement of the trunk and limbs. Each functional class is further subdivided into subtypes exhibiting distinct physiological, morphological, and molecular properties (*Lallemend and Ernfors, 2012*; *Usoskin et al., 2015*). Since the same Neurogenin1-dependent progenitor population gives rise to the three principal DRG neuron types within a relatively short time window (*Frank and Sanes, 1991*; *Ma et al., 1999*), specification of the diverse DRG neuron subtypes mainly takes place in postmitotic neurons. Therefore, DRG sensory neurons are particularly well suited for mechanistic studies of postmitotic differentiation of neuronal subtypes.

The differentiation of nociceptor precursors into nonpeptidergic and peptidergic nociceptors is a well-studied example of postmitotic sensory neuron subtype specification. These two nociceptor subtypes are molecularly, morphologically and functionally distinct. Peptidergic nociceptors are traditionally defined by expression of neuropeptides, such as CGRP and Substance P, while the majority of nonpeptidergic nociceptors have binding sites for the lectin IB4 (*Mulderry et al., 1988*; *Silverman and Kruger, 1990*). More recently, a single cell transcriptome analysis led to the identification of four molecularly defined nonpeptidergic neuron subtypes (*Usoskin et al., 2015*). Peptidergic and nonpeptidergic nociceptor subtypes are further distinguished by their central and peripheral axonal projection patterns (*Molliver et al., 1995*; *Zylka et al., 2005*). Functionally, nonpeptidergic and peptidergic nociceptors are preferentially required for mechanical and thermal nociception, respectively (*Cavanaugh et al., 2009*; *McCoy et al., 2013*; *Mishra and Hoon, 2010*; *Vulchanova et al., 2001*).

Despite these differences, nonpeptidergic and peptidergic nociceptors derive from a common nociceptor precursor that expresses tropomyosin-receptor kinase A (TrkA), the receptor for nerve

growth factor (NGF), and both nociceptor subtypes require NGF–TrkA signaling for survival and development beginning at ~E12.5 (*Crowley et al., 1994*; *Luo et al., 2007*; *Patel et al., 2000*; *Silos-Santiago et al., 1995*; *Smeyne et al., 1994*). A key developmental event that defines the segregation of nociceptor subtypes is the nonpeptidergic nociceptor-specific switch from NGF responsiveness to glial-derived neurotrophic factor (GDNF) family ligand (GFL) responsiveness. This occurs by virtue of the upregulation of Ret and GDNF family *receptor alpha receptors* (GFRα), receptor components for GFL signaling, in nonpeptidergic nociceptors starting from E15.5, as well as postnatal extinction of TrkA expression (*Bennett et al., 1996*; *Bennett et al., 1998*; *Molliver and Snider, 1997*; *Molliver et al., 1997*). Interestingly, both NGF–TrkA and GFL–GFRα/Ret signaling are essential for postmitotic differentiation of nonpeptidergic nociceptors. Indeed, NGF–TrkA signaling is required for the acquisition of virtually all nonpeptidergic nociceptor-specific features, including expression of *Ret* and *Gfras*, whereas GFL–GFRα/Ret signaling, in turn, plays a critical role in postnatal development of nonpeptidergic nociceptors, including expression of a subset of genes characteristic of mature nonpeptidergic nociceptors (*Luo et al., 2007*; *Patel et al., 2000*). The mechanism by which NGF–TrkA signaling governs differentiation of nonpeptidergic nociceptors is unclear.

In addition to the extrinsic signals NGF and GFLs, intrinsic determinants of nonpeptidergic nociceptor fate have begun to be defined (*Chen et al., 2006*; *Lopes et al., 2012*). Most prominent is Runx1, a Runx family transcription factor whose expression is primarily restricted to nonpeptidergic nociceptors. Runx1 functions as a master regulator of the nonpeptidergic nociceptor lineage (*Chen et al., 2006*; *Yoshikawa et al., 2007*). In mammals, Runx family proteins, which include Runx1, Runx2 and Runx3, are key regulators of hematopoietic, osteogenic, and immune cell lineages (*Banerjee et al., 1997*; *de Bruijn and Speck, 2004*; *Ducy et al., 1997*). In those systems, Runx proteins function as heterodimers by forming complexes with CBFβ, a non-DNA-binding cofactor that enhances the DNA binding affinity and protein stability of Runx proteins (*Adya et al., 2000*). In the peripheral nervous system, genetic ablation of Runx1 leads to a near complete loss of nonpeptidergic nociceptor-specific features and a concomitant expansion of neurons exhibiting a peptidergic nociceptor-like phenotype (*Chen et al., 2006*; *Kramer et al., 2006*; *Yoshikawa et al., 2007*). Interestingly, the dramatic consequences of NGF deficiency on nonpeptidergic-specific gene expression resemble those seen upon Runx1 ablation (*Chen et al., 2006*; *Luo et al., 2007*). This observation suggests a potential interaction between the extrinsic signal NGF and the intrinsic factor Runx1 during nonpeptidergic nociceptor development and thus an opportunity to define mechanisms of interplay between extrinsic and intrinsic factors during postmitotic neuronal differentiation.

Here, we show that Runx1 controls nonpeptidergic nociceptor development primarily by acting downstream of NGF–TrkA signaling. NGF/TrkA regulates Runx1 function at least in part by promoting expression of CBFβ, which we show is an essential component of the Runx1/CBFβ complex during nonpeptidergic nociceptor development. Mechanistically, NGF–TrkA signaling controls Runx1/CBFβ complex formation through the ERK/MAPK signaling pathway, which is both necessary and sufficient for *Cbfb* expression. On the other hand, Islet1, a LIM-homeodomain transcription factor, controls transcriptional initiation of *Runx1* but not *Cbfb*. These findings identify a novel NGF/TrkA–MAPK–CBFβ axis critical for the differentiation of nonpeptidergic nociceptors and reveal a mechanism by which convergence of extrinsic and intrinsic signals promotes formation of a heterodimeric transcription factor complex that instructs postmitotic neuronal subtype specification.

## Results

### NGF and Runx1 are similarly required for initiation of nonpeptidergic nociceptor-specific gene expression

The differentiation of nonpeptidergic nociceptors requires the target-derived neurotrophic growth factor NGF as well as the transcription factor Runx1 (*Chen et al., 2006*; *Luo et al., 2007*). The phenotypic similarity exhibited by mice lacking NGF and those lacking Runx1 raises the intriguing possibility that NGF and Runx1 function in a common signaling pathway to control nonpeptidergic nociceptor maturation. To address this, we assessed the extent to which initiation of the nonpeptidergic phenotype is co-dependent on NGF and Runx1 during early stages of nociceptor development. We first sought to extend our understanding of Runx1-dependent nonpeptidergic nociceptor-specific genes using unbiased gene expression profiling of DRGs from *Wnt1-Cre;*

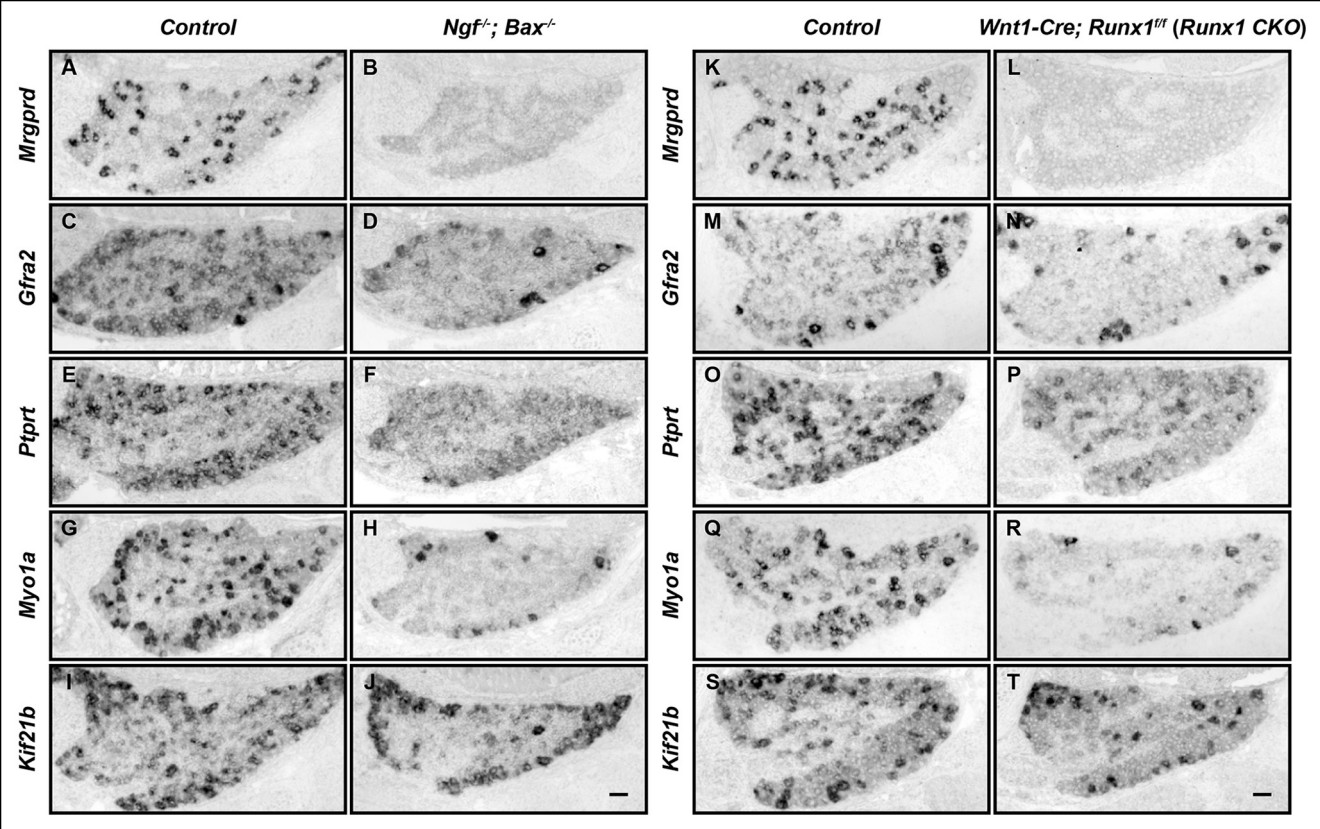

**Figure 1.** The majority of nonpeptidergic nociceptor-specific genes depend on both NGF and Runx1 for expression. (A–J) Expression of *Mrgprd* (Control, 12.7% ± 2.4%; *Ngf*[-/-]*Bax*[-/-], 0%), *Gfra2* (Control, 28.7% ± 2.5%; *Ngf*[-/-]*Bax*[-/-], 14.3% ± 1.1%), *Ptprt* (Control, 19.1% ± 0.3%; *Ngf*[-/-]*Bax*[-/-], 10.3% ± 2.1%), *Myo1a* (Control, 26.4% ± 0.9%; *Ngf*[-/-]*Bax*[-/-], 5.8% ± 1.5%) and *Kif21b* (Control, 27.2% ± 2.7%; *Ngf*[-/-]*Bax*[-/-], 14.05% ± 1.8%) in control and *Ngf*[-/-]*Bax*[-/-] DRGs at P0, assessed by in situ hybridization. (K–T) Expression of *Mrgprd* (Control, 26.3% ± 1.0%; *Runx1 CKO*, 0.3% ± 0.2%), *Gfra2* (Control, 40.0% ± 3.7%; *Runx1 CKO*, 9.6% ± 0.7%), *Ptprt* (Control, 36.0% ± 3.3%; *Runx1 CKO*, 10.4% ± 2.3%), *Myo1a* (Control, 31.1% ± 2.3%; *Runx1 CKO*, 5.1% ± 1.8%) and *Kif21b* (Control, 24.8% ± 1.7%; *Runx1 CKO*, 10.9% ± 0.8%) in control and *Runx1 CKO* DRGs at P0, assessed by in situ hybridization. Note that *Ngf*[-/-]*Bax*[-/-] and *Runx1 CKO* mutants display similar deficits in expression of these representative nonpeptidergic-nociceptor specific genes. For *Gfra2, Myo1a, Kif21b*, the apparent NGF- and Runx1-independent expression in large diameter neurons can be attributed to their expression in non-nociceptive DRG neurons. Shown are the means ± SEM for the percentage of DRG neurons expressing indicated genes based on counts from a total of at least 9 sections from three independent animals per genotype. DRG neurons were identified and counted based on combined NeuN immunostaining, which was not shown. *Ngf*[+/-]; *Bax*[-/-] or *Ngf*[+/+];*Bax*[-/-] and *Runx1*[f/f] mice were used as control animals for analysis of *Ngf*[-/-];*Bax*[-/-] and *Runx1 CKO* mutants, respectively. Scale bar, 50 μm. See also *Figure 1—figure supplement 1*.

The following figure supplements are available for Figure 1:

**Figure supplement 1.** The majority of nonpeptidergic nociceptor-specific genes depend on both NGF and Runx1 for initiation of expression.

*Runx1*[f/f] *(Runx1 CKO)* mice, previously characterized neural crest-specific *Runx1* conditional mutants and their littermate controls at E16.5, the onset of nonpeptidergic-specific gene expression (*Supplementary file 1*) (*Chen et al., 2006*). Among the many genes with downregulated expression in *Runx1CKO* DRGs compared to controls, *Ptprt, Myo1a*, and *Kif21b* were shown to exhibit strong Runx1 dependence by in situ hybridization analysis (*Figure 1O–T*). Further co-localization studies with Runx1 confirmed that their expression is primarily restricted to nonpeptidergic nociceptors (data not shown). We next compared the patterns of expression of these as well as additional, canonical nonpeptidergic nociceptor-specific genes between *Runx1 CKO* mice and mice lacking NGF at the same developmental stages by in situ hybridization. In order to study survival-independent functions of NGF, nociceptors are kept alive in the absence of NGF by co-deletion of *Ngf* and the proapoptotic gene *Bax* (*Ngf*[-/-]; *Bax*[-/-] mice) (*Patel et al., 2000*). In accordance with previous observations, expression of the nonpeptidergic nociceptor markers *Mrgprd* and *Gfra2* was

almost completely abolished in presumptive nonpeptidergic nociceptors of both *Ngf$^{-/-}$; Bax$^{-/-}$* and *Runx1 CKO* mice at P0 (*Figure 1A–D, K–N*) (*Chen et al., 2006*; *Luo et al., 2007*). Moreover, newly identified nonpeptidergic-specific genes, including *Ptprt, Myo1a,* and *Kif21b,* exhibited dramatically reduced expression in neurons that would normally become nonpeptidergic nociceptors in *Ngf$^{-/-}$; Bax$^{-/-}$* mice compared to littermate controls at P0 (*Figure 1E–J*), in a manner similar to that observed in *Runx1 CKO* mice (*Figure 1O–T*). Furthermore, impairment in expression of *Mrgprd, Ptprt, Myo1a,* and *Kif21b* was observed in *Ngf$^{-/-}$; Bax$^{-/-}$* and *Runx1 CKO* mice as early as E16.5, which is when their expression is normally initiated (*Figure 1—figure supplement 1*). As expected, expression of *Gfra2, Myo1a,* and *Kif21b* in non-nociceptors was unaffected in *Ngf$^{-/-}$; Bax$^{-/-}$* and *Runx1 CKO* mice, thus explaining the residual expression observed in these mutants (*Figure 1D, N* and *Figure 1—figure supplement 1F, H, N, P*). These results together demonstrate a general requirement of both NGF and Runx1 for initiation of expression of a large subset of nonpeptidergic neuron genes.

While the large majority of nonpeptidergic-specific genes require both NGF and Runx1 for initiation of expression, expression of the nonpeptidergic marker *Ret* displays differential dependence on NGF and Runx1. Although *Ret* expression is severely impaired in *Ngf$^{-/-}$; Bax$^{-/-}$* DRGs at P0, as previously described (*Figure 2A–C*) (*Luo et al., 2007*), its expression at this time point is only mildly affected in *Runx1 CKO* DRGs compared to controls, as shown by both in situ hybridization and real-time PCR (*Figure 2F, G, J*). A time-course analysis of *Ret* expression in control and *Runx1 CKO* DRGs using both in situ hybridization and real-time PCR further confirmed the perinatal onset of Runx1 dependence for *Ret* expression (*Figure 2D–J*). Therefore, while NGF is required for initiation of *Ret* expression, Runx1 is only required at a later stage, for maintenance of *Ret* expression. We conclude that NGF and Runx1 are both required for initiation of expression of a large cohort of nonpeptidergic nociceptor-specific genes, with *Ret* being a notable exception.

## Runx1 acts downstream of NGF signaling to control expression of nonpeptidergic nociceptor-specific genes

The finding that initiation of expression of most nonpeptidergic nociceptor-specific genes analyzed so far is dependent on both NGF and Runx1 suggests a model in which Runx1 is a downstream mediator of NGF signaling during maturation of nonpeptidergic nociceptors. Alternatively, Runx1 may indirectly control nonpeptidergic nociceptor maturation by facilitating NGF signaling. In fact, the level of NGF signaling, as assessed by the fluorescent intensity of phospho-Trk (pTrk) immunostaining, is lower in *Runx1 CKO* DRG neurons relative to controls at P0 (*Figure 3—figure supplement 1A–F*). This ability of Runx1 to sustain normal levels of TrkA activity is not simply due to an effect on TrkA expression, as TrkA levels in control and *Runx1 CKO* DRGs are indistinguishable, as determined by immunohistochemistry (*Figure 3—figure supplement 1G, H*). However, since *Ret* expression, which is strongly dependent on NGF signaling in nonpeptidergic nociceptors, appears normal at E16.5 (*Figure 2D, E*), it is unlikely that diminished NGF signaling observed at P0 accounts for the profound deficits in initiation of expression of nonpeptidergic-specific genes in E16.5 *Runx1 CKO* mice (*Figure 1—figure supplement 1*). Nevertheless, to directly test the possibility that Runx1 controls maturation of nonpeptidergic nociceptors solely by modulating NGF signaling, we asked whether exogenous NGF can rescue nonpeptidergic nociceptor gene expression deficits in *Runx1 CKO* DRGs. This was addressed in two ways. First, the consequence of excess NGF was evaluated in vivo by in situ hybridization and real-time PCR analysis following intraperitoneal injections of NGF into neonatal mice. With the notable exception of *Ret*, whose expression was partially restored following NGF injection into *Runx1 CKO* mice (*Figure 3J–L* and *Figure 3—figure supplement 2D*), we found a general and complete lack of effect of NGF administration on expression of nonpeptidergic-specific genes in *Runx1 CKO* DRGs (*Figure 3A–I* and *Figure 3—figure supplement 2A–C*). In a second series of experiments, dissociated DRG cultures from P0 control and *Runx1 CKO* animals were incubated in the presence or absence of NGF, and the effects of NGF on nonpeptidergic-specific gene expression were assessed by real-time PCR. Importantly, NGF treatment led to a robust increase in expression of the nonpeptidergic-specific genes, *Mrgprd, Gfra2,* and *Ptprt,* in wildtype neurons but not in Runx1-deficient neurons (*Figure 2M–O*). In contrast, *Ret* expression was NGF-dependent in both control and Runx1-deficient neurons, albeit to a lesser extent in the absence of Runx1 (*Figure 3P*). Taken together, these findings are most consistent with a model in which Runx1

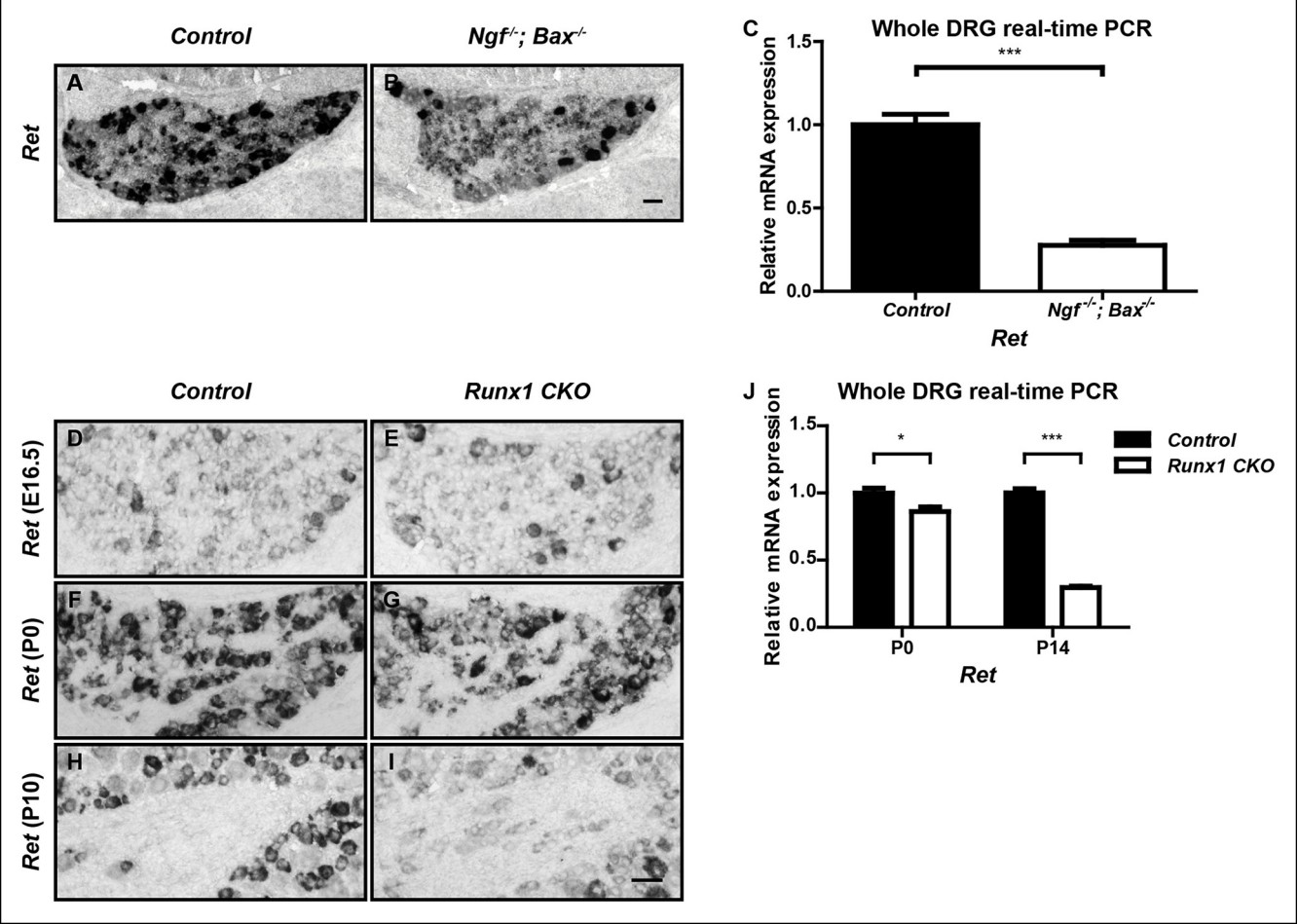

**Figure 2.** *Ret* is an unusual nonpeptidergic nociceptor-specific gene whose expression is differentially dependent on NGF and Runx1. (**A and B**) Greatly diminished *Ret* expression in *Ngf*[-/-]*Bax*[-/-] DRGs compared to controls at P0, assessed by in situ hybridization. The NGF-independent Ret[+] neurons are mechanoreceptors (Aβ RA-LTMRs). (**C**) Real-time PCR analysis of *Ret* expression in control and *Ngf*[-/-]*Bax*[-/-] DRGs at P0 confirms its strong NGF dependence in small diameter neurons. Statistical analysis was done using an unpaired t test, N = 4, ***p ≤ 0.001. (**D–I**) *Ret* expression in control and *Runx1 CKO* DRGs at E16.5 (**D** and **E**), P0 (**F** and **G**) and P10 (**H** and **I**), assessed by in situ hybridization. Note that while *Ret* expression is almost completely eliminated in *Ngf*[-/-]*Bax*[-/-] DRGs at P0, its expression at this same time point in *Runx1 CKO* DRGs is only modestly affected, indicating different temporal requirements of NGF and Runx1 for *Ret* expression. Shown are representative results of at least two independent animals per genotype at each time point. (**J**) Real-time PCR analysis of *Ret* expression in control and *Runx1 CKO* DRGs at P0 and P14 confirms the progressive nature of the *Ret* expression deficit in *Runx1 CKO* DRGs. Statistical analyses were done using unpaired t tests, N = 3 for each time point, *p ≤ 0.05, ***p ≤ 0.001. *Ngf*[+/-]; *Bax*[-/-] or *Ngf*[+/+]; *Bax*[-/-] and *Runx1*[f/f] mice were used as control animals for analysis of *Ngf*[-/-]; *Bax*[-/-] and *Runx1 CKO* mutants, respectively. Scale bar, 50 μm.

primarily acts downstream of NGF to control the initiation of expression of nonpeptidergic-specific genes.

## Runx1 and CBFβ form a complex in DRG neurons

To further test this model, we explored the possibility that NGF promotes Runx1 activity during nonpeptidergic nociceptor maturation. To address the mechanism by which Runx1 activity may be modulated by NGF, it was critical to first ask how Runx1 activity is normally controlled in DRG neurons. Outside of the nervous system, Runx family transcription factors function as heterodimers, forming complexes with a common non-DNA-binding cofactor CBFβ. CBFβ is indispensable for Runx1 activity in cells of the hematopoietic lineage because of its role in enhancing the DNA-binding activity and stability of Runx1 proteins (*Adya et al., 2000*). Therefore, we asked whether CBFβ plays a similar role in DRG neurons to promote Runx1 function.

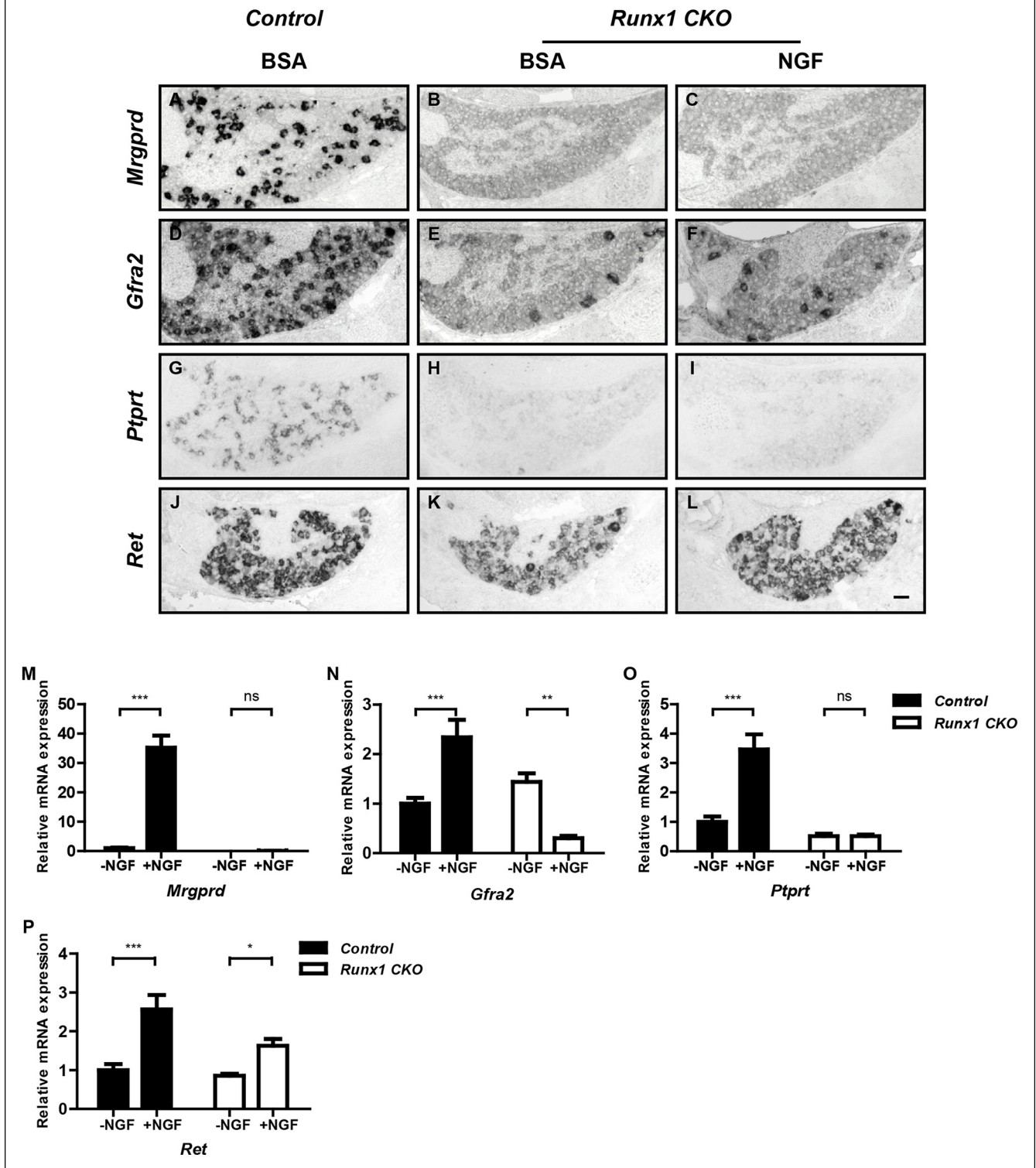

**Figure 3.** Runx1 functions downstream of NGF to mediate expression of the majority of nonpeptidergic nociceptor-specific genes, whereas it controls *Ret* expression at least in part by enhancing NGF signaling. (A–L) In situ hybridization analysis of expression of *Mrgprd* (A–C), *Gfra2* (D–F), *Ptprt* (G–I) and *Ret* (J–L) in DRGs of P2 control animals that received BSA injections, *Runx1CKO* animals that received BSA injections or *Runx1 CKO* animals that received NGF injections. Note that exogenous NGF administration fails to activate expression of nonpeptidergic-specific genes in *Runx1 CKO* animals, with the notable exception of *Ret*, suggesting that Runx1 is a downstream mediator of NGF signaling that is required for expression of the majority of nonpeptidergic-specific genes. The ability of exogenous NGF to upregulate *Ret* expression in *Runx1 CKO* animals is most consistent with an indirect

*Figure 3. continued on next page*

*Figure 3. Continued*

role for Runx1 in regulating *Ret* expression, through enabling NGF signaling. Shown are results representative of at least three independent injection experiments. See also *Figure 3—figure supplements 1, 2*. (M–P) Real-time PCR analysis of expression of *Mrgprd* (M), *Gfra2* (N), *Ptprt* (O) and *Ret* (P) in dissociated DRG neurons from P0 control and *Runx1 CKO* animals cultured in the presence or absence of NGF. Note that, with the exception of *Ret*, NGF-dependent expression of these nonpeptidergic-specific genes is completely abolished in the absence of Runx1, further supporting Runx1 as a downstream mediator of NGF in regulating expression of most nonpeptidergic-specific genes. Statistical analyses were done using two-way ANOVA with a Bonferroni post-test, N = 5 for M and P, N = 7 for the rest. *p ≤ 0.05, **p ≤ 0.01, ***p ≤ 0.001, ns non-significant. *Runx1^{f/f}* mice were used as control animals for analysis of *Runx1 CKO* mutants. Scale bar, 50 µm.

The following figure supplements are available for Figure 3:

**Figure supplement 1.** Runx1 potentiates TrkA activity without regulating TrkA expression.

**Figure supplement 2.** Runx1 controls expression of the majority of nonpeptidergic-specific genes independent of its stimulatory effect on NGF signaling.

To determine whether CBFβ is expressed in DRG neurons at the appropriate time to form a complex with Runx1, its expression patterns at both the mRNA and protein levels during nociceptor development were assessed. In situ hybridization analysis revealed that *Cbfb* is expressed at varying levels, in the majority, if not all, DRG neurons over the entire time course of our analysis (*Figure 4—figure supplement 1B–D*). Due to a lack of CBFβ antibodies that work reliably for immunohistochemistry, we generated a *Cbfb^{Flag}* knockin mouse line, in which N-terminally Flag-tagged CBFβ protein (Flag-CBFβ) is produced from the endogenous *Cbfb* locus (*Figure 4—figure supplement 1A*). In *Cbfb^{Flag}* mice, the Flag antibody allows for specific detection of endogenous Flag-CBFβ, which appears to be expressed in nearly all DRG neurons, at varying levels (*Figure 4A, B*). Double labeling of Flag and Runx1 showed overlap between CBFβ and Runx1 throughout nociceptor development suggesting a potential interaction between these two proteins (*Figure 4C–E* and *Figure 4—figure supplement 1E–G*). It is noteworthy that CBFβ is broadly expressed and found in more than just Runx1^{+} neurons, suggesting additional Runx1-independent CBFβ functions in the DRG (*Figure 4C–E*). To test for a direct physical interaction between Runx1 and CBFβ in DRG neurons, co-immunoprecipitation experiments were done using a Flag antibody and extracts from DRGs of P0 *Cbfb^{Flag/Flag}* animals and wildtype littermate controls. Indeed, Runx1 is enriched in Flag immunoprecipitates from *Cbfb^{Flag/Flag}* animals but not wildtype animals (*Figure 4F*). Thus, Runx1 and CBFβ are co-expressed and form a complex in developing DRG neurons.

## CBFβ is required for the development of Runx1-dependent nonpeptidergic populations

To assess the function of CBFβ in DRG development and, in particular, the role of CBFβ in Runx1-dependent nonpeptidergic nociceptor maturation, which has not previously been feasible because of the early lethality of *Cbfb* null embryos, we next generated a conditional *Cbfb* allele (*Cbfb^f*) by flanking the putative promoter sequence and the first two exons of this gene with *LoxP* sites (*Figure 5—figure supplement 1A*). Effective gene ablation in the DRG was achieved by crossing mice harboring the *Cbfb^f* allele to a *Wnt1-Cre* mouse line, which drives recombination specifically in the dorsal neural tube and neural crest derivatives (Danielian et al., 1998) (*Figure 5—figure supplement 1B, C*). Analysis of *Wnt1-Cre; Cbfb^{f/f}* (*Cbfb CKO*) animals revealed a wide range of nonpeptidergic nociceptor phenotypes that are virtually identical to those seen in *Runx1 CKO* mice. Specifically, those genes found to be Runx1-dependent similarly require CBFβ for normal expression, as shown by in situ hybridization and real-time PCR analysis at P0 (*Figure 5A–J* and *Figure 5—figure supplement 1D, E*). As has been shown for *Runx1 CKO* animals, expression of CGRP, a marker of peptidergic nociceptors, was de-repressed in *Cbfb CKO* animals (data not shown). Furthermore, like *Runx1 CKO* mice, *Cbfb CKO* animals at P0 exhibited a marked reduction of sensory innervation of the epidermis (*Figure 5K–P*). This defect was primarily due to a deficiency in the peripheral projections of nonpeptidergic nociceptors, considering the sparseness of epidermal innervation by peptidergic nociceptors at P0 as well as their unperturbed innervation pattern in the absence of CBFβ (data not shown). We also found that the subepidermal nerve plexus is unaffected in *Cbfb CKO* mice,

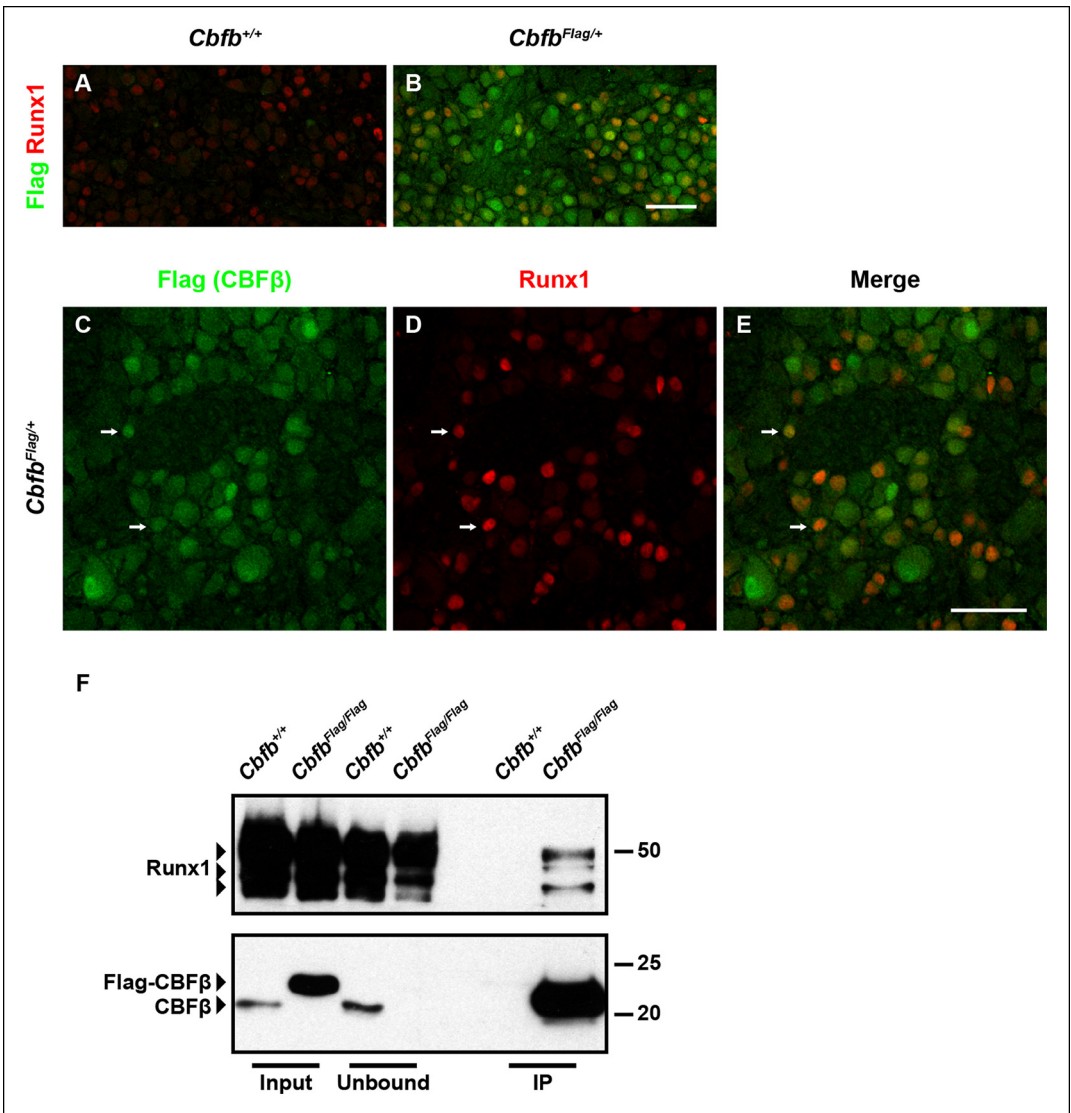

**Figure 4.** Runx1 and CBFβ are co-expressed and form a complex in DRG neurons. (**A** and **B**) Double immunostaining of Flag and Runx1 in wildtype and *Cbfb*[Flag/+] DRGs at P0 confirms the specificity of the Flag antibody. Note that Flag immunoreactivity in wildtype DRGs is nearly undetectable. (**C–E**) Double immunostaining of Flag and Runx1 in *Cbfb*[Flag/+] DRGs at P0 shows extensive colocalization between Flag-CBFβ and Runx1 (arrows). Note that CBFβ is expressed in many more DRG neurons than those that are Runx[1+]. See more examples in *Figure 4—figure supplement 1E–G*. (**F**) Co-immunoprecipitation experiments using a Flag antibody for immunoprecipitation from DRGs lysates from P0 wildtype and *Cbfb*[Flag/Flag] animals. Subsequent detection with Runx1 and CBFβ antibodies shows that Runx1 associates with Flag-CBFβ from DRGs of *Cbfb*[Flag/Flag] animals, but not wildtype controls, indicating the formation of a Runx1/CBFβ complex in the DRG. Scale bar, 50 μm.

The following figure supplements are available for Figure 4:

**Figure supplement 1.** Generation of the *Cbfb*[Flag] allele and more detailed characterization of the temporal and spatial patterns of *Cbfb* expression.

indicating a developmental defect in the final stage of peripheral target innervation. Therefore, both molecular and morphological features of nonpeptidergic nociceptors critically depend on both Runx1 and CBFβ during embryonic development.

We next asked whether CBFβ is required at postnatal times for development of nonpeptidergic nociceptors. Since *Wnt1Cre; Cbfb CKO* animals die perinatally due to craniofacial defects, a

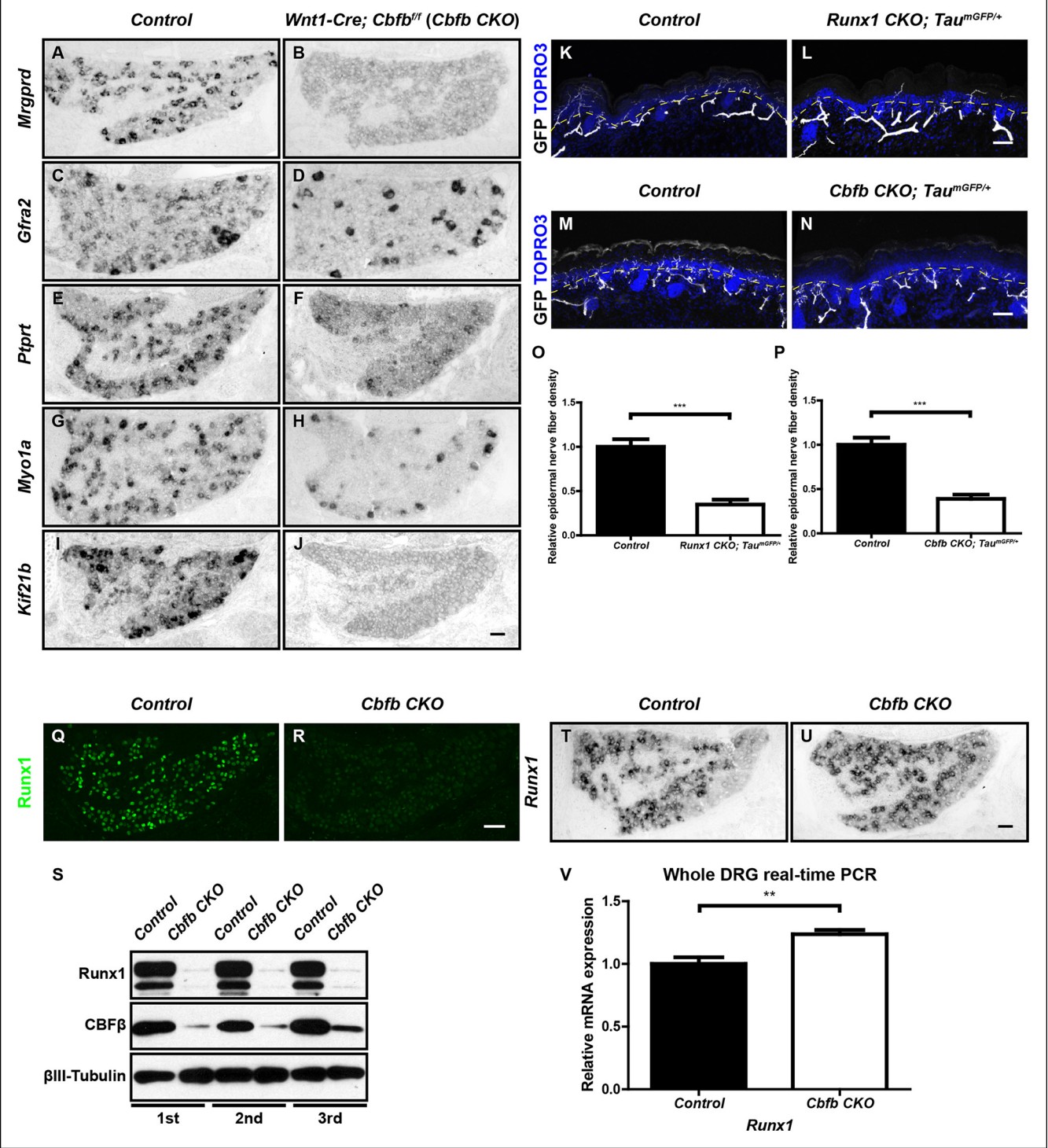

**Figure 5.** CBFβ is required for acquisition of molecular and morphological features of nonpeptidergic nociceptors. (A–J) Expression of *Mrgprd* (Control, 26.9% ± 2.8%; *Cbfb CKO*, 0%), *Gfra2* (Control, 38.8% ± 2.8%; *Cbfb CKO*, 11.7% ± 1.9%), *Ptprt*, (Control, 31.9% ± 3.2%; *Cbfb CKO*, 7.1% ± 2.8%), *Myo1a* (Control, 26.9% ± 3.2%; *Cbfb CKO*, 5.6% ± 0.6%) and *Kif21b* (Control, 20.2% ± 0.1%; *Cbfb CKO*, 2.4% ± 0.5%) in control and *Cbfb CKO* DRGs at P0 by in situ hybridization analysis. The gene expression deficits in *Cbfb CKO* animals phenocopy those observed in *Runx1 CKO* animals except for *Kif21b* expression. The discrepancy likely reflects *Kif21b* expression in proprioceptors where it presumably depends on Runx3 and CBFβ for expression. Shown are the means ± SEMs for the percentage of neurons expressing indicated genes based on counts from a total of at least 9 sections from three independent animals per genotype. DRG neurons were identified and counted based on combined NeuN immunostaining, which was not shown. See also *Figure 5—figure supplement 1D, E*. (K–N) GFP immunostaining of P0 hairy skin to visualize sensory innervation of the epidermis in control and *Runx1 CKO* animals (K and L) or control and *Cbfb CKO* animals (M and N) that also carry the *Tau^{mGFP}* allele. The *Tau^{mGFP}* allele was

*Figure 5. Continued*

introduced to label all Cre-expressing neurons including all sensory neurons. Note that there is a dramatic reduction in fiber density specifically in the epidermis in both *Runx1 CKO* and *Cfb CKO* animals relative to their littermate controls. The yellow dotted line denotes the epidermal-dermal junction which was drawn based on TOPRO3 counterstain (blue). (**O and P**) Quantification of sensory innervation of the epidermis in control and *Runx1 CKO* animals (**O**) or control and *Cfb CKO* animals (**P**) reveals a remarkably similar reduction in the innervation density in both mutants at P0. The innervation density is defined as the fraction of area occupied by GFP+ fibers in the epidermis. An unpaired t test was performed on data from three independent animals per genotype. ***p ≤ 0.001. (**Q and R**) Runx1 immunostaining of control and *Cfb CKO* DRGs at P0 shows almost complete loss of Runx1 proteins in the absence of CBFβ. Shown are representative images from at least three independent experiments. (**S**) Immunoblot analysis of expression of *Runx1* and *Cfb* in control and *Cfb CKO* DRGs at P0 shows dramatic loss of Runx1 proteins as a result of CBFβ depletion. βIII-Tubulin serves as a loading control. Shown are results from three independent experiments. (**T and U**) In situ hybridization analysis of *Runx1* expression in control and *Cfb CKO* DRGs at P0 shows comparable levels of *Runx1* transcripts in control and mutant animals. (**V**) Real-time PCR analysis of *Runx1* expression in control and *Cfb CKO* DRGs at P0 shows increased *Runx1* mRNA expression in *Cfb CKO* DRGs compared to control, which likely reflects an increased ratio of nociceptors to proprioceptors (data not shown). An unpaired t test was performed on data from four independent pairs of control and mutant animals, **p ≤ 0.01. *Cfb$^{f/f}$* mice were used as control animals for analysis of *Cfb CKO* mutants. Scale bar, 50 μm.

The following figure supplements are available for Figure 5:

**Figure supplement 1.** Generation of the *Cfb$^f$* allele and demonstration of a postnatal requirement for both CBFβ and Runx1 in C-LTMR development.

*Runx1$^{CreER}$* knockin allele (*Samokhvalov et al., 2007*) combined with postnatal tamoxifen administration was used to ablate *Cfb* postnatally. A similar strategy was used to generate a *Runx1* inducible knockout mouse model for direct comparison. Both *Runx1$^{CreER}$; Cfb$^{f/f}$* and *Runx1$^{CreER/f}$* mice treated with postnatal tamoxifen were viable and indistinguishable from control littermates. However, following postnatal deletion of either *Runx1* or *Cfb*, we observed, during the third postnatal week, few or no tyrosine hydroxylase (TH)+ C-low threshold mechanoreceptors (C-LTMRs), a nonpeptidergic neuronal subtype (*Li et al., 2011*; *Seal et al., 2009*). Specifically, key features of C-LTMRs, including TH expression and longitudinal lanceolate endings in the periphery, as marked by expression of a Cre-dependent GFP reporter (*Hippenmeyer et al., 2005*), are nearly completely absent in mice lacking Runx1 or CBFβ at postnatal time points (*Figure 5—figure supplement 1F–M*). Similar phenotypes were previously reported in a different *Runx1* conditional mutant (*Lou et al., 2013*). Therefore, both Runx1 and CBFβ are required during early development for initiation of the nonpeptidergic nociceptor fate and at postnatal times for maturation of at least one specific nonpeptidergic neuronal subtype, the C-LTMR.

The phenocopy of *Runx1* and *Cfb* mutants may be partly attributed to a dramatic defect in Runx1 protein expression in *Cfb CKO* DRGs, as shown by immunostaining and western blot analysis at P0 (*Figure 5Q–S*). This Runx1 protein deficit was evident as early as E13 (data not shown). *Runx1* mRNA expression, on the other hand, remained unchanged, if not increased, in *Cfb CKO* DRGs compared to controls, as determined by both in situ hybridization and real-time PCR (*Figure 5T–V*). These findings demonstrate a key role for CBFβ in the post-transcriptional regulation of *Runx1* expression, most likely at the level of protein stability. Together, these findings indicate that CBFβ and Runx1 are both essential for induction of the nonpeptidergic nociceptor fate and postnatal nonpeptidergic neuron subtype maturation, and that the CBFβ requirement may reflect its role in controlling the level and activity of Runx1 proteins.

## NGF regulates expression of *Cfb* but not *Runx1* at early stages of nociceptor maturation

How does NGF control Runx1/CBFβ activity? Since Runx1 and CBFβ are both necessary for nonpeptidergic neuron maturation, NGF could, in principle, control Runx1/CBFβ function by promoting the activity of Runx1, CBFβ or both. Considering the profound requirement of NGF for nonpeptidergic nociceptor gene expression, one possibility is that NGF promotes expression of *Runx1*, *Cfb*, or both, thereby enabling Runx1/CBFβ complex formation, stabilization and function, and thus initiation of the nonpeptidergic nociceptor fate. We therefore examined expression of both *Cfb* and *Runx1* in DRGs of control and *Ngf$^{-/-}$; Bax$^{-/-}$* animals before and during the acquisition of the nonpeptidergic nociceptor fate. Interestingly, *Cfb* expression was found to be sensitive to loss of NGF prior to specification of nonpeptidergic nociceptors. In fact, at E14.5, a time prior to expression of nonpeptidergic nociceptor-specific genes, *Cfb* mRNA expression was significantly reduced in *Ngf$^{-/-}$; Bax$^{-/-}$*

DRGs relative to controls, specifically in small-diameter nociceptor precursors (*Figure 6A, B, E*). This *Cbfb* expression deficit in *Ngf*[-/-]; *Bax*[-/-] small diameter DRG neurons becomes much more pronounced at later time stages, such as E16.5, the onset of observable deficits in nonpeptidergic-specific gene expression (*Figure 6C–E*). Similarly, *Ntrk1*[-/-]/*Bax*[-/-] DRGsdisplayed reduced levels of *Cbfb* expression at P0 (*Figure 7H, I*). The deficit of *Cbfb* mRNA expression was confirmed by real-time PCR analysis (*Figure 6—figure supplement 1A*). Considering the inability of this assay to distinguish between *Cbfb* expressed in nociceptors and that expressed in NGF-independent DRG neurons, such as proprioceptors, and non-neuronal cells of the ganglion, it is noteworthy that the real-time PCR measurements are an underestimation of the NGF dependence of *Cbfb* expression in developing nociceptors. We further addressed NGF-dependence of *Cbfb* expression at the protein level in two ways. First, Flag immunostaining on dissociated DRG cultures from P0 *Cbfb*[Flag/+] animals showed that NGF is essential for CBFβ protein expression in vitro (*Figure 6F–H*). Second, when DRGs from P0 control and *Ngf*[-/-]; *Bax*[-/-] animals that were also heterozygous for the *Cbfb*[Flag] allele were acutely dissociated, the level of Flag immunoreactivity was significantly lower in *Ngf*[-/-]; *Bax*[-/-] neurons compared to controls, suggesting NGF-dependence of CBFβ protein expression in vivo (*Figure 6—figure supplement 1B–D*). *Cbfb* is also sensitive to the dose of NGF as exogenous NGF administered to wildtype neonates via intraperitoneal injection further potentiated *Cbfb* expression (*Figure 6—figure supplement 1E*). Together, these findings identify NGF as a key regulator of *Cbfb* expression prior to the initiation of NGF-dependent nonpeptidergic-specific gene expression. On the other hand, *Runx1* mRNA levels are normal in *Ngf*[-/-]; *Bax*[-/-] DRGs at E14.5, as determined by both in situ hybridization and real-time PCR analysis (*Figure 6I, J, M*), consistent with our previous observation (*Luo et al., 2007*). It is only at later times, beginning at E16.5, that *Runx1* expression becomes affected, suggesting a late requirement of NGF for maintenance of *Runx1* expression (*Figure 6K–M*). Consistent with the late NGF dependence of *Runx1* mRNA expression, Runx1 protein expression is unaffected in *Ngf*[-/-]; *Bax*[-/-] DRGs at E14.5 (*Figure 6N, O, R, S*). At E16.5, the level of Runx1 protein is significantly diminished without any change in the number of Runx1[+] neurons (*Figure 6P–S*). In view of the discrepancy between the dramatic nonpeptidergic phenotypes at E16.5 and the relatively modest deficit in *Runx1* expression at this time, upregulation of *Cbfb* expression by NGF is likely to be an important mechanism by which NGF enables Runx1 function during nonpeptidergic nociceptor development.

## NGF stimulates *Cbfb* expression in a MAPK-dependent manner

To better understand the mechanism by which NGF controls *Cbfb* expression in nociceptor precursors, we next sought to identify NGF–TrkA signaling cascades that control *Cbfb* expression. The canonical ERK1/2 signaling cascade represents a likely candidate because this is a major NGF–TrkA effector pathway, and animals deficient in components of this pathway exhibit nonpeptidergic phenotypes, including reduced *Ret* expression, impaired innervation of the epidermis, and impaired adult CBFβ protein expression (*Newbern et al., 2011*; *Zhong et al., 2007*). To directly assess the role of MAPK signaling in NGF-dependent *Cbfb* expression during development, both in vitro and in vivo approaches were used. Through immunostaining and immunoblot analysis, we found that pharmacological inactivation of Mek1/2, direct activators of ERK1/2, greatly attenuated the ability of NGF to promote CBFβ protein expression in vitro (*Figure 7A–E* and *Figure 7—figure supplement 1A–C*). A *Nestin-Cre*-based conditional knockout mouse model, which targets all four *Mapk3*, *Mapk1*, *Map2k1*and *Map2k2* alleles, was next used to determine the in vivo role of MAPK signaling for *Cbfb* expression. Using this in vivo model system, we found that in 4 out of 5 P0 *Nes-Cre*; *Map2k1*[f/f;]*Map2k2*[-/-]; *Mapk3*[-/-]; *Mapk1*[f/f] mutants (*Quadruple*), *Cbfb* mRNA expression was severely disrupted, demonstrating a strong dependence of *Cbfb* expression on MAPK signaling in vivo (*Figure 7F, G*). The phenotypic variation in this analysis likely reflects incomplete excision of all four floxed alleles. We next asked whether MAPK signaling is sufficient to promote *Cbfb* expression in vivo, in the absence of NGF or activation of other NGF-dependent signaling pathways. For this analysis, a constitutively active form of B-Raf (V600E) was expressed exclusively in the nervous system to drive MAPK signaling in animals that were null for both *Ntrk1* and *Bax* (*O'Donovan et al., 2014*). Remarkably, while *Cbfb* expression in *Ntrk1*[-/-]; *Bax*[-/-] DRGs was severely impaired, expression of B-Raf[V600E] in *Ntrk1*[-/-]; *Bax*[-/-] mice to promote constitutive MAPK signaling restored *Cbfb* expression to near normal levels (*Figure 7H–J*). Real-time PCR analysis was carried out as an independent measure of *Cbfb* expression to further confirm the necessity and sufficiency of MAPK signaling for *Cbfb*

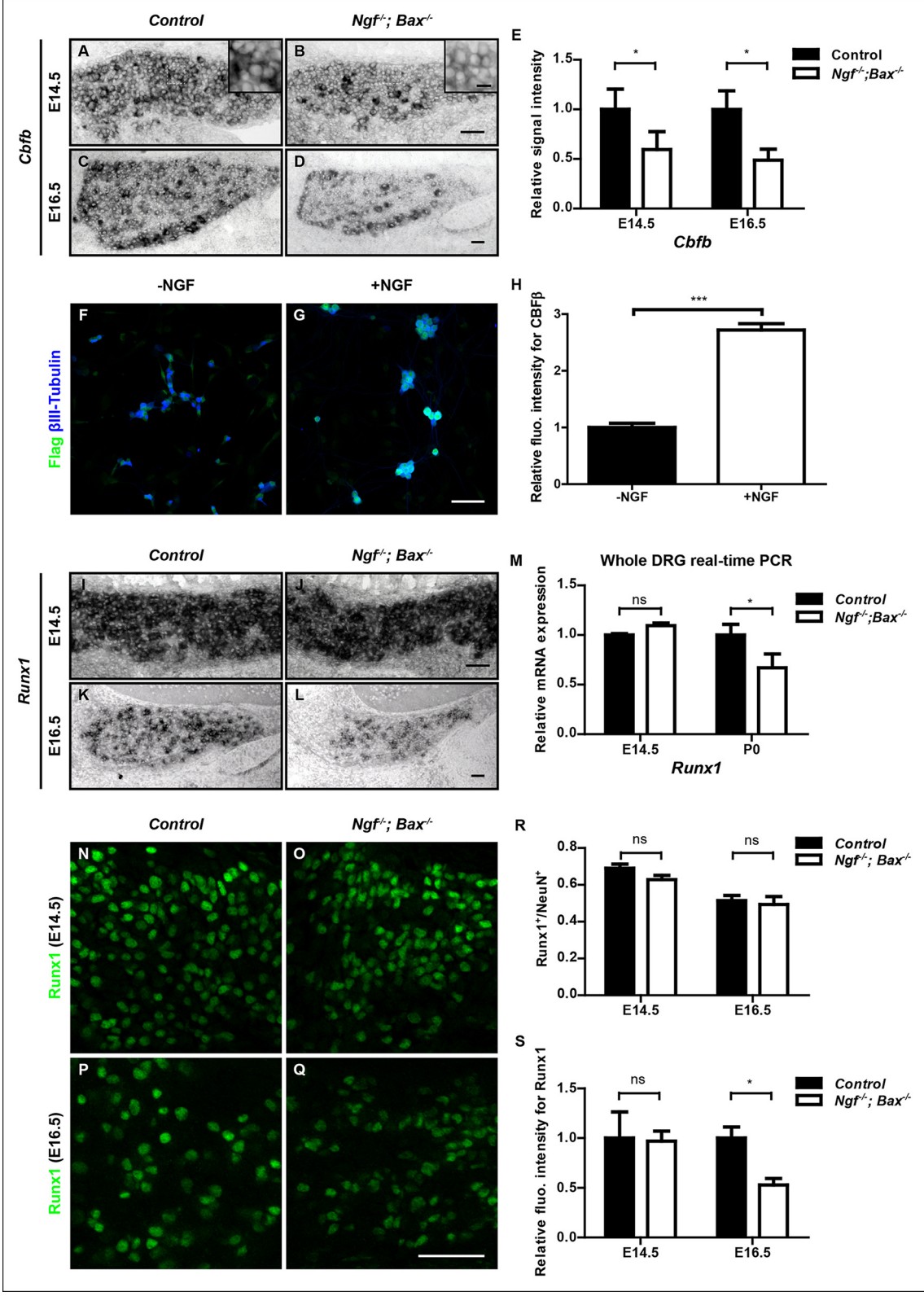

**Figure 6.** NGF regulates the Runx1/CBFβ complex through differential control of *Cbfb* and *Runx1* expression. (**A and B**) In situ hybridization analysis of *Cbfb* expression in control and *Ngf*[-/-]*Bax*[-/-] DRGs at E14.5 shows a significant reduction in the level of transcripts in small diameter neurons that correspond to prospective nociceptors in *Ngf*[-/-]*Bax*[-/-] DRGs compared to controls. The insets focus on nociceptor-rich regions. Scale bar for the insets, 10μm. Note that *Cbfb* in situ hybridization was combined with Runx3 immunostaining to exclude the Runx3[+] *Cbfb* population from the analysis. (**C and**
*Figure 6. continued on next page*

Figure 6. Continued

D) In situ hybridization analysis of *Cbfb* expression in control and *Ngf⁻ᐟ⁻Bax⁻ᐟ⁻* DRGs at E16.5 shows more pronounced *Cbfb* mRNA deficit in *Ngf⁻ᐟ⁻Bax⁻ᐟ⁻* DRGs. (E) Quantification of *Cbfb* expression deficits in nociceptors in *Ngf⁻ᐟ⁻Bax⁻ᐟ⁻* DRGs based on experiments described in (A–D). Intensity of in situ signal in areas devoid of Runx3⁺ neurons was measured. An unpaired t test was performed using data collected from 3 independent experiments for each time point. *$p \leq 0.05$. See also *Figure 6—figure supplement 1A*. (F and G) Double staining of Flag and βIII-Tubulin in dissociated DRG neurons from P0 *Cbfb^{Flag/+}* animals that were cultured without or with NGF. Note that NGF application robustly stimulates CBFβ protein expression as indicated by increased Flag immunoreactivity. (H) Quantification of the effect of NGF treatment on CBFβ protein levels based on experiments described in (F and G). CBFβ protein abundance was quantified using the average fluorescence intensity of Flag immunoreactivity per cell. An unpaired t test was performed using data collected from four independent experiments. ***$p < 0.0001$. See also *Figure 6—figure supplement 1B–D*. (I and J) In situ hybridization analysis of *Runx1* expression in control and *Ngf⁻ᐟ⁻Bax⁻ᐟ⁻* DRGs at E14.5 shows comparable levels of *Runx1* transcripts in control and mutant DRGs. Means ± SEM for relative intensity of in situ signals after normalization to the level in control DRGs is as follows: Control, 1.00 ± 0.16; *Ngf⁻ᐟ⁻Bax⁻ᐟ⁻*, 0.73 ± 0.12. $p = 0.2079$, based on an unpaired t test. (K and L) In situ hybridization analysis of *Runx1* expression in control and *Ngf⁻ᐟ⁻Bax⁻ᐟ⁻* DRGs at E16.5 shows a reduction in the level of signal per cell in *Ngf⁻ᐟ⁻Bax⁻ᐟ⁻* DRGs compared to controls. Control, 1.00 ± 0.07; *Ngf⁻ᐟ⁻Bax⁻ᐟ⁻*, 0.49 ± 0.06. $p = 0.0003$, based on an unpaired t test. (M) Real-time PCR analysis of *Runx1* expression in control and *Ngf⁻ᐟ⁻Bax⁻ᐟ⁻* DRGs at E14.5 and P0 reveals a late requirement of NGF for *Runx1* expression. An unpaired t test was performed on data collected from three independent animals per genotype at each time point. *$p \leq 0.05$, ns non-significant. (N–Q) Runx1 immunostaining in control and *Ngf⁻ᐟ⁻Bax⁻ᐟ⁻* DRGs at E14.5 (N and O) and E16.5 (P and Q) shows that the Runx1 protein deficit becomes evident in *Ngf⁻ᐟ⁻Bax⁻ᐟ⁻* DRGs at E16.5, which coincides with the onset of nonpeptidergic nociceptor deficits in *Ngf⁻ᐟ⁻Bax⁻ᐟ⁻* DRGs. (R and S) Quantification of Runx1 protein expression in control and *Ngf⁻ᐟ⁻Bax⁻ᐟ⁻* DRGs at E14.5 and E16.5 based on the percentage of Runx1⁺ neurons or the fluorescence intensity of Runx1 immunoreactivity. Note that loss of NGF specifically affects the level of Runx1 expression per cell without altering the number of Runx1⁺ neurons. An unpaired t test was performed on data collected from three independent animals per genotype. *$p \leq 0.05$, ns non-significant. *Ngf⁺ᐟ⁻; Bax⁻ᐟ⁻* or *Ngf⁺ᐟ⁺; Bax⁻ᐟ⁻* mice were used as control animals for analysis of *Ngf⁻ᐟ⁻; Bax⁻ᐟ⁻* mutants. Scale bar, 50 μm.

The following figure supplements are available for Figure 6:

**Figure supplement 1.** *Cbfb* expression is NGF-dependent in vivo.

expression in vivo (*Figure 7K*). Together, these findings indicate that the ERK/MAPK signaling cascade is both necessary and sufficient to mediate NGF/TrkA-dependent expression of *Cbfb*.

## Islet1, a LIM-homeodomain transcription factor, is required for initiation of expression of *Runx1* but not *Cbfb*

While *Cbfb* expression at early stages of nonpeptidergic nociceptor development is dependent on NGF, the apparent NGF-independence of *Runx1* expression during early development prompted us to ask whether initiation of *Cbfb* and *Runx1* expression are differentially controlled by intrinsic cues. We tested the involvement of Islet1, a LIM-homeodomain transcription factor, because in a neural crest derivative-specific *Islet1* mutant (*Isl1 CKO*), a Runx1 protein deficit was noted as early as E12.5, the time when Runx1 proteins are first detected in lumbar DRGs (*Dykes et al., 2011*; *Sun et al., 2008*). To define the level at which *Runx1* expression is regulated by Islet1, *Runx1* expression was evaluated by in situ hybridization analysis in DRGs of E12.5 control and *Isl1 CKO* animals. Consistent with a central role for Islet in initiating *Runx1* expression at the transcriptional level, *Runx1* transcripts were virtually undetectable in *Isl1 CKO* DRGs (*Figure 8A, B*). In contrast, *Cbfb* expression was only minimally affected by the same genetic perturbation (*Figure 8C, D*). The distinct dependence of *Runx1* and *Cbfb* expression on Islet1 was confirmed by a microarray analysis of control and *Isl1 CKO* DRGs at E12.5 (*Figure 8E*). Thus, expression of Runx1 and CBFβ, obligatory components of a transcription factor holocomplex, are under differential control of Islet1 and NGF; *Runx1* and *Cbfb* at early stages of nonpeptidergic nociceptor differentiation are preferentially regulated by the intrinsic cue Islet1 and the extrinsic cue NGF, respectively. Thus, a convergence of intrinsic and extrinsic signaling events in nonpeptidergic nociceptor progenitors enables formation of the Runx1/CBFβ transcription factor complex, a key event required for nonpeptidergic nociceptor differentiation.

## Discussion

This study defines a gene regulatory mechanism underlying the specification of nonpeptidergic nociceptors. At the core of this process is the formation of the Runx1/CBFβ transcription factor complex, both components of which are essential for directing the nonpeptidergic nociceptor-specific

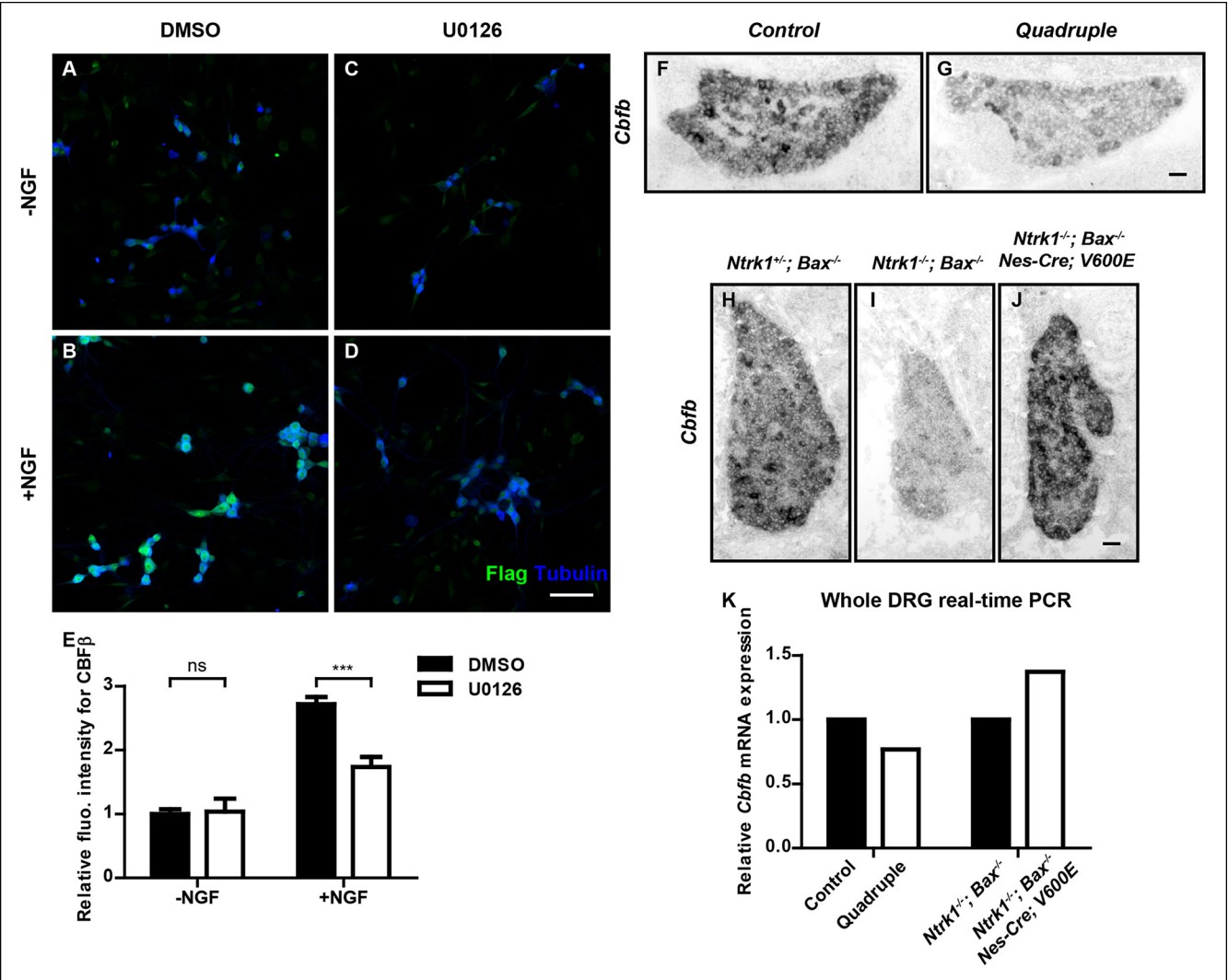

**Figure 7.** NGF promotes *Cbfb* expression through the ERK/MAPK signaling pathway. (**A–D**) Double staining of Flag (green) and βIII-Tubulin (blue) in DMSO or U0126-treated dissociated DRG neurons from P0 *Cbfb^{Flag/+}* animals that were cultured without or with NGF. U0126 is a selective inhibitor of MEK1/2, the direct activators of ERK1/2. Note that CBFβ protein expression, as defined by Flag immunoreactivity is greatly diminished in U0126-treated neurons as compared to vehicle-treated neurons, all grown in the presence of NGF. (**E**) Quantification of the effect of U0126 treatment on CBFβ protein levels based on experiments as described in (**A–D**). CBFβ protein abundance was quantified as the average fluorescence intensity of Flag immunoreactivity per cell. Statistical analysis was done using a two-way ANOVA with a Bonferroni post-test, based on data from four independent experiments. ***p ≤ 0.001, ns non-significant. See also *Figure 7—figure supplement 1*. (**F and G**) In situ hybridization analysis of *Cbfb* expression in control and *quadruple* DRGs at P0 reveals a severe deficit in *Cbfb* mRNA expression in DRGs when MAPK signaling is disrupted in the nervous system. A similar phenotype of varied severity was observed in 4 out of 5 mutant animals. (**H–J**) In situ hybridization analysis of *Cbfb* expression in *Ntrk1^{+/−}; Bax^{−/−}, Ntrk1^{−/−}; Bax^{−/−}* and *Ntrk1^{−/−}; Bax^{−/−}; Nes-Cre; V600E* DRGs at E18.5 shows that constitutive activation of MAPK signaling leads to a dramatic increase in *Cbfb* expression in TrkA-deficient animals. Shown are representative images from two independent experiments. (**K**) Real-time PCR analysis of *Cbfb* expression in the same set of loss-of-function and gain-of-function mouse models as described in (**F and G**) and (**H–J**) at P0 further demonstrates the necessity and sufficiency of MAPK signaling for NGF-dependent *Cbfb* expression in vivo. Shown are averages from two independent experiments after normalization to littermate control. Scale bar, 50 μm.

The following figure supplements are available for Figure 7:

**Figure supplement 1.** In vitro evidence for the necessity of MAPK signaling for NGF-dependent CBFβ expression.

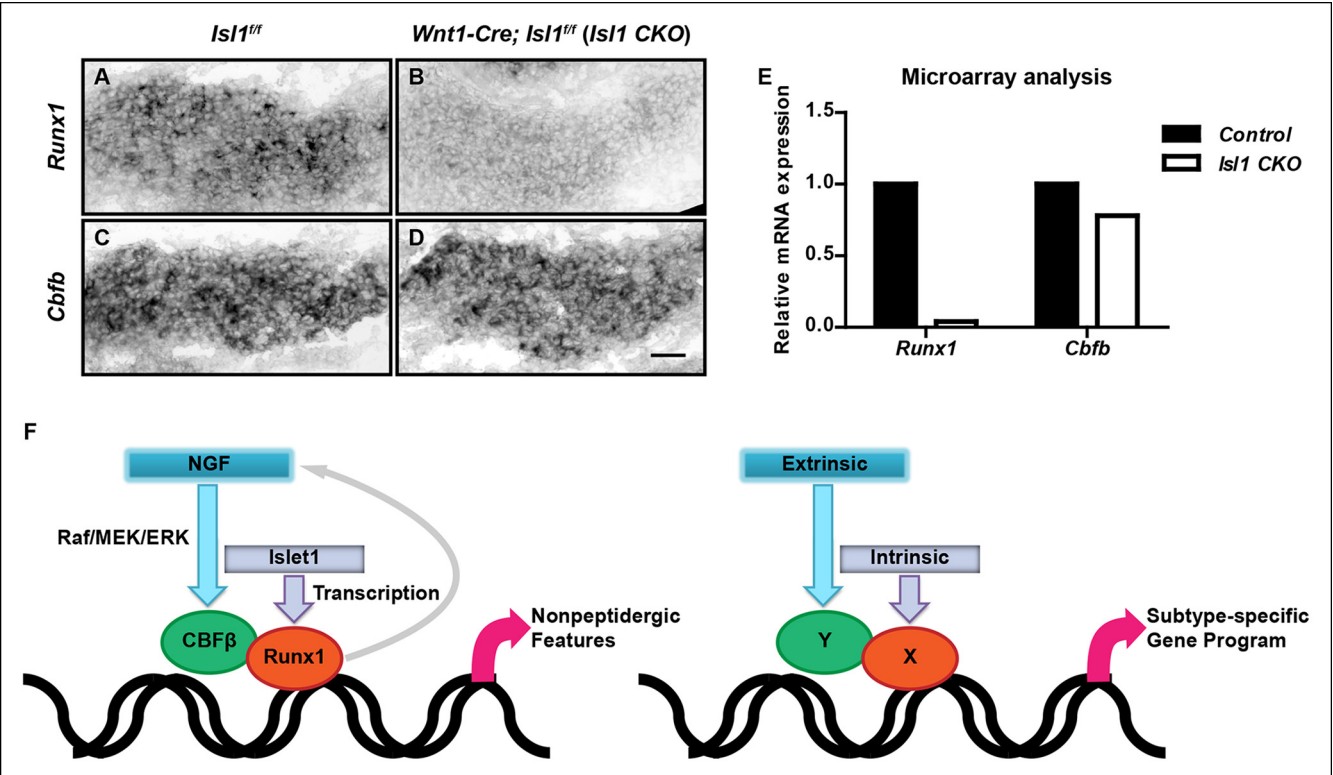

**Figure 8.** Islet1 is required for initiation of *Runx1*, but not *Cbfb* expression. (**A–D**) In situ hybridization analysis of expression of *Runx1* (**A** and **B**) and *Cbfb* (**C** and **D**) in control and *Isl1 CKO* DRGs at E12.5 shows that *Islet1* deficiency abolishes expression of *Runx1* but not *Cbfb* at an early age. Shown are representative images from two independent experiments. (**E**) Microarray analysis of E12.5 control and *Isl1 CKO* DRGs further confirms the differential dependence of expression of *Runx1* and *Cbfb* on Islet1. Shown are average expression levels from two independent experiments that are normalized to the control levels for each gene. Expression levels have been normalized using globe scaling. *Isl1*<sup>f/f</sup> mice were used as control animals for analysis of *Isl1 CKO* mutants. (**F**) Schematics illustrating a molecular mechanism underlying specification of nonpeptidergic nociceptors and its general implication in the context of subtype specification. The extrinsic cue NGF and the intrinsic cue Islet1 coordinately regulate the Runx1/CBFβ complex, a nonpeptidergic nociceptor transcription factor complex, by preferentially targeting *Cbfb* and *Runx1* for transcriptional regulation, respectively. Furthermore, the Runx1/CBFβ complex, through an unknown mechanism, enhances the level of NGF-TrkA signaling, resulting in a positive feedback loop between NGF-TrkA signaling and Runx1/CBFβ complex. This gene regulatory mechanism not only underscores the importance of interplay between extrinsic and intrinsic factors during multilineage differentiation, but also illustrates how such interplay can control cell-fate decisions through the convergence of extrinsic and intrinsic signals at the level of a heterodimeric, lineage-specific transcription factor complex. Scale bar, 50 μm.

transcriptional program. Importantly, expression of each component is independently regulated at the transcriptional level. *Cbfb* expression is controlled by the extrinsic cue NGF through the MAPK signaling pathway. On the other hand, *Runx1* expression is initiated by a mechanism that is critically dependent on Islet1, an intrinsic factor that controls early development of sensory neurons. Furthermore, the Runx1/CBFβ complex, through an unknown mechanism, maintains a high level of NGF-TrkA signaling, which is essential for at least one defining feature of nonpeptidergic nociceptors: postnatal expression of *Ret*. Thus, the initiation of a lineage-specific differentiation program is tightly controlled by a convergence of extrinsic and intrinsic factors at the level of formation of the heterodimeric Runx1/CBFβ transcription factor complex (*Figure 8F*).

## The Runx1/CBFβ complex, a coincidence detector for extrinsic and intrinsic cues that direct nociceptor subtype specification

Nonpeptidergic nociceptor specification requires proper expression of both *Runx1* and *Cbfb*, whose transcriptional initiation is differentially dependent on Islet1 and NGF signaling. The Runx1/CBFβ holocomplex therefore serves as a coincidence detector for extrinsic and intrinsic signals that promote specification of nonpeptidergic nociceptors. The requirement for NGF, an extrinsic cue that is critical for survival of all nociceptors, ensures that only those nociceptor precursors that gain access

to a sufficient amount of NGF and survive the period of naturally occurring cell death will undergo nonpeptidergic nociceptor maturation. Consistent with this, *Cbfb* expression exhibits NGF dependence at E14.5, immediately following the period of naturally occurring cell death. On the other hand, the dependence of *Runx1* expression on Islet1, a transcription factor required for terminating expression of genes whose expression is restricted to the early stage of sensory neurogenesis, likely contributes to the orderly transition from pan-sensory neurogenesis to specification of nonpeptidergic nociceptors by ensuring that *Runx1* expression and hence Runx1/CBFβ-dependent nonpeptidergic nociceptor development are initiated after sensory neurogenesis (*Sun et al., 2008*). Thus the NGF and Islet1 signals coordinate temporal control of the Runx1/CBFβ complex for timely initiation of nonpeptidergic nociceptor differentiation. It is important to note that the extrinsic and intrinsic signals described here are unlikely to be sufficient to instruct Runx1/CBFβ complex formation and the nonpeptidergic nociceptor lineage choice, as neither NGF nor Islet1 functions exclusively in nonpeptidergic nociceptors. Indeed, NGF–TrkA signaling is required for survival, target innervation and normal phenotypic development of both peptidergic and nonpeptidergic nociceptors (*Harrington and Ginty, 2013*), and Islet1 is expressed broadly in developing DRG neurons and is required for the generation of virtually all nociceptors (*Sun et al., 2008*). Therefore, one or more additional, unidentified signals must govern the divergence of the two main nociceptive lineages.

## Transcriptional activator(s) of *Cbfb* expression downstream of MAPK signaling

We found that MAPK signaling is both necessary and sufficient for NGF-dependent expression of *Cbfb* in developing nociceptors. The generality of MAPK signaling suggests that it plays a similar role in controlling *Cbfb* expression in cells outside of the nervous system and thus MAPK–CBFβ signaling may have a general role in cell type specification. Moreover, the identification of MAPK as an upstream activator of *Cbfb* expression may provide an explanation for the observation that *Cbfb*, unlike *Runx* genes, is widely expressed (*Wang et al., 1993*). The identity of nuclear targets of the MAPK signaling pathway that directly activate *Cbfb* expression remains unclear. To this end, we found, through bioinformatic analysis of the putative *Cbfb* promoter, cAMP-response element (CRE) consensus motifs within an evolutionarily conserved 458 bp enhancer-like element (data not shown), raising the intriguing possibility that CREB family members, which are well-studied downstream mediators of MAPK signaling in many cell types (*Shaywitz and Greenberg, 1999*), are direct transcriptional activators of *Cbfb*. In view of the role of CREB family members in NGF-dependent growth and survival of neurons (*Liu et al., 1999*; *Riccio et al., 1999*), the possibility of CREB-mediated activation of *Cbfb* expression would further suggest that common effectors of growth factor signaling pathways support distinct biological outcomes, in this case, survival, axon growth and maturation of nociceptors, presumably by controlling expression of distinct target genes.

## A mechanism of interplay between intrinsic and extrinsic signals for postmitotic differentiation of neuronal subtypes

The formation of the Runx1/CBFβ complex downstream of NGF and Islet1 signaling during nonpeptidergic nociceptor maturation illustrates a novel mechanism of interplay between extrinsic and intrinsic factors in controlling postmitotic specification of neuronal subtypes. It is important to note the distinction between this relatively late developmental process and the specification of progenitor domains, which takes place prior to cell cycle exit. While it is well established in vertebrate systems that specification of progenitor domains is coordinately regulated by the intrinsically defined competence state of a progenitor and spatially and temporally controlled extrinsic signals (*Briscoe and Novitch, 2008*; *Livesey and Cepko, 2001*; *Molyneaux et al., 2007*), relatively little is known about the contributions of extrinsic cues, intrinsic factors, and their modes of interaction, during postmitotic specification of neuronal subtypes. The gene regulatory mechanism described here leads us to propose a simple model in which convergence of extrinsic and intrinsic signals onto a single heterodimeric transcription factor complex controls lineage-specific differentiation programs and postmitotic specification of neuronal subtypes (*Figure 8F*). It is conceivable that this simple model is generally applicable to a broad range of neuronal subtypes that rely on heterodimeric transcription factor complexes for their specification.

# Materials and methods

## Mice

For generation of mice harboring the *Cbfb^f* allele, a 2 kb sequence containing a 1 kb sequence immediately upstream of the transcription start site as well as exon 1 and exon 2 of the *Cbfb* locus was flanked by two *loxP* sites using recombineering technology. For generation of mice harboring the *Cbfb^Flag* allele, the sequence encoding a single Flag epitope was introduced immediately upstream of the translational start site of the *Cbfb* gene. Mice were generated using targeted ES cells and standard blastocyst injection techniques. *Cbfb^f/f* mice were mated to either a mouse strain expressing Cre recombinase under control of the *Wnt1* promoter (*Danielian et al., 1998*) to generate *Wnt1-Cre; Cbfb^f/f* mice or to a mouse strain in which the Cre recombinase coding sequence was inserted into the *Runx1* locus (*Samokhvalov et al., 2007*) to generate *Runx1^CreER/+; Cbfb^f/f* mice. *Wnt1-Cre; Runx1^f/f* and *Ngf^−/−; Bax^−/−* mice were generated as described (*Chen et al., 2006*; *Patel et al., 2000*). The mouse lines used to generate *Nes-Cre; Map2k1^f/f; Map2k2^-/-; Mapk3^-/-; Mapk1^f/f* and *Ntrk1^-/-; Bax^-/-; Nes-Cre; V600E* mice were described previously (*Mercer et al., 2005*; *Moqrich et al., 2004*; *Newbern et al., 2011*). The *Tau^mGFP* allele, a neuronal specific Cre-dependent GFP reporter, was previously described (*Hippenmeyer et al., 2005*). See supplemental information for details on generation of the *Cbfb^f* and *Cbfb^Flag* alleles.

## In situ hybridization and immunohistochemistry

Digoxigenin (DIG)-labeled *cRNA* probes were used for in situ hybridization. Target sequences for *Ptprt, Myo1a, Kif21b* probes were amplified using gene specific PCR primers from either cDNA prepared from P0 mouse DRGs or genomic DNA from wildtype ES cells to generate corresponding plasmids for in situ hybridization. In situ hybridization probes for *Mrgprd, Gfra2, Ret* and *Runx1* were described previously (*Luo et al., 2007*). The in situ hybridization probe for *Cbfb* was generated from an available cDNA clone (GenBank: BC026749.1). In situ hybridizations were carried out on 14 μm fresh frozen DRG sections as described previously (*Luo et al., 2007*). For combined in situ hybridization and immunostaining, regular BCIP/NBT-based in situ hybridization was performed prior to the standard immunostaining procedure. Bright field and fluorescent images were taken under the same setting. Bright field images were later pseudocolored and merged with fluorescent images.

Immunohistochemistry on DRG and skin sections was performed as described previously (*Li et al., 2011*; *Luo et al., 2007*). The primary antibodies used were: rabbit anti-Runx1 (a gift from Dr. Thomas Jessell, Columbia University, 1:10,000), rabbit anti-Runx3 (a gift from Dr. Thomas Jessell, Columbia University, 1:4000), guinea pig anti-Flag (see supplemental information for antibody generation details, 1:500), rabbit anti-CGRP (Immunostar, 24112, 1:1000), chicken anti-GFP (Aves Labs, GFP-1020, 1:500), chicken anti-NF200 (Aves Labs, NFH, 1:500), rabbit anti-Tyrosine Hydroxylase (Millipore, AB152, 1:1000), rabbit anti-pTrk-SHC (Cell Signaling Technology, 4168,1:500), rabbit anti-pTrk-PLCγ (Cell Signaling Technology, 4619,1:500), rabbit anti-TrkA (Millipore, AB1577, 1:1000), mouse anti-NeuN (Milllipore, MAB377MI, 1:500), and rabbit anti-βIII-Tubulin (Covance, PRB-435P, 1:1000).

## Dissociated DRG neuronal cultures

Dissociated DRG cultures from neonatal mice were prepared using a method that was adapted from a previously described protocol for sympathetic neuronal cultures (*Deckwerth et al., 1996*). Briefly, neurons were obtained by sequential steps of enzymatic digestion and mechanical dissociation of DRGs harvested from P0 animals. In general, these neurons were plated on Poly-D-lysine and laminin coated coverslips at a density of ~50,000 neurons per well and cultured for 2 days in growth media (10% FBS, 1 U/ml penicillin/streptomycin, 5 μm Ara-C (Sigma), 50 μg/ml a pan-caspase inhibitor Boc-aspartyl (OMe)-fluoromethylketone (BAF) (MP Biomedicals)) supplemented with NGF (100 ng/ml; either purified from mouse salivary glands or purchased from Millipore), or a neutralizing NGF antibody (Sigma) at a dilution of 1:2000. For experiments that used U0126, cultures were treated with U0126 (50 μm in DMSO; Calbiochem) or DMSO the morning after plating. For experiments that addressed in vivo dependence of CBFβ protein expression on NGF, neurons were plated for 1 hour in growth media without supplements before being processed for Flag immunostaining.

## Real-time PCR

RNA was extracted from freshly isolated DRGs using the RNeasy micro kit (Qiagen) according to the manufacturer's instructions. First strand cDNA was synthesized using an oligo dT primer and the SuperScript III system (Invitrogen). Real-time PCR was performed using the QuantiTect SYBR Green PCR kit (Qiagen) and a 7300 Real-Time PCR System (Applied Biosystems). The amount of individual transcripts was normalized to that of *PGP9.5*, a pan-neuronal marker, unless the comparison was between control and *Ngf*[−/−]; *Bax*[−/−], for which *GAPDH* served as the internal control. Detailed primer sequences for real-time PCR can be found in supplemental information.

## Immunoblotting and co-immunoprecipitation

Acutely dissected DRGs were lysed in ice-cold FA-M2 Lysis Buffer (50 mM Tris HCl, 150 mM NaCl, 1 mM EDTA, 1% Triton X-100, pH 7.5) supplemented with a protease inhibitor cocktail (Sigma;1:100 dilution) by sonication. Clarified lysates were either subjected to Flag immunoprecipitation for co-immunoprecipitation experiments or processed directly for SDS-PAGE. Immunoprecipitation of Flag-CBFβ and its associated proteins was done using anti-Flag M2 affinity gel (Sigma) according to the manufacturer's instructions. Immunoblotting was performed using antibodies against Runx1 (Abcam, 1:5000), CBFβ (1:1000, Santa Cruz), Histone 3 (1:1000, Cell Signaling Technology) and βIII-Tubulin (1:1000, Covance), as described (*Kuruvilla et al., 2000*).

## Statistical analysis

Statistical differences for mean values between two groups and among multiple groups were analyzed using GraphPad Prism 5 software. The type of test used for statistical analysis is indicated in the figure legend. The criterion for statistical significance was set at $p \leq 0.05$.

## Mouse line generation

For the *Cbfb*[f] allele, a ~2 kb sequence (chromosome 8: 105169674- 105171592) corresponding to a 1 kb sequence immediately upstream of the transcriptional start site as well as exon 1 and exon 2 of the *Cbfb* locus was flanked by two *loxP* sites. Recombineering technology was used to generate the targeting vector (*Copeland et al., 2001*; *Liu et al., 2003*). Briefly, a 129/SvJ BAC clone containing the targeted region of the *Cbfb* gene was obtained from Geneservice. An 11.5 kb region (chromosome 8: 105168174-105179387) with homology arms that were 1.5 kb and 8 kb long was inserted into a PBS-DTA plasmid, the backbone for the final targeting vector, via the first recombineering step. The 3' *loxP* site and the *FRT-Neo-FRT*-5' *loxP* cassette were then introduced sequentially during subsequent recombineering steps. A Bstz171 restriction site was engineered 3' to the 3' *loxP* site to facilitate southern screening of ES cells. The targeting construct was linearized with KpnI and then electroporated into mouse 129S6SvEvTac ES cells. ES clones were screened by PCR and correctly targeted ES clones were confirmed by southern blot hybridization using both 5' and internal probes following Bstz171 digestion (WT 9.8 kb and Mutant 6.8 kb, data not shown). Chimeric *Cbfb*[f] mice were produced by injection of positive ES cells into C57Bl/6 blastocysts. Mice carrying the *Cbfb*[f] allele were subsequently generated by mating chimeric mice to germ-line *FlpE* mice to remove the *Neo* cassette (*Rodriguez et al., 2000*). *Cbfb*[f] mice were genotyped using a 2-primer PCR reaction with the following primers: 5'-GCGCGCCAGTCACTTGTT-3' and 5'-AAACCATCCCAC-GAACCGAACCAT-3'. The sizes of PCR products from wildtype and mutant alleles are 219 bp and 324 bp, respectively. For the *Cbfb*[Flag] allele, the targeting vector, which was nearly identical to that of the *Cbfb*[f] vector, was generated using a combination of recombineering and standard subcloning strategies. The same targeted genomic region was engineered to include the *FRT-Neo-FRT-loxP* cassette at the position identical to that in the *Cbfb*[f] allele, using recombineering technology. The sequence encoding one Flag epitope was introduced into the vector immediately upstream of the translational start site of the *Cbfb* gene by replacing a 1.1 kb NotI/AvrII fragment containing the translational start site with the fragment carrying the insertion using standard cloning techniques. A Bstz171 restriction site was inserted immediately downstream of the *Flag* sequence for the purpose of southern screening. Subsequent steps for generation of *Cbfb*[Flag] mice were the same as those described above for *Cbfb*[f] mice. *Cbfb*[Flag] mice were genotyped using a 2-primer PCR reaction with the following primers: 5'-TGAGAGCTGTCTATGGCAAAC-3' and 5'-

TCAGTTCAAGGATGGCAGGTA-3'. The sizes of PCR products from wildtype and mutant alleles are 232 bp and 336 bp, respectively.

## Real-time PCR
Primers used for real-time PCR analysis are provided *Table 1*.

## NGF administration via intraperitoneal injections
Mouse pups of the desired genotype were subjected to a single intraperitoneal injection of either NGF (2 μg reconstituted in PBS with 1% BSA) or equal volume of 1% BSA in PBS at both P0 and P1. Animals were sacrificed at P2 and vertebral columns or DRGs were dissected and processed for analysis.

## Tamoxifen injections
Tamoxifen (Toronto Research Chemicals) was dissolved in ethanol (20 mg/ml). 50 μl (1 mg) of tamoxifen in ethanol was mixed with 50 μl of sunflower seed oil (Sigma), vortexed for 20 min and centrifuged under vacuum for 45 min to remove the ethanol. The tamoxifen solution was delivered at P2 via an intraperitoneal injection into animals harboring the *Runx1*$^{CreER}$ allele.

## Microarray analysis
A total of 6 RNA samples (~1 μg each) were prepared using Trizol and the RNeasy micro kit from DRGs of three pairs of E16.5 control and *Runx1 CKO* animals, each from different litters. Samples were labeled and hybridized to Affymetrix mouse 430 2.0 chips and microarray data were analyzed with Spotfire software. Only genes with a fold change greater than or equal to 1.5, a p-value less than or equal to 0.05 were considered differentially expressed between control and *Runx1 CKO* DRGs and were reported. Methods for the microarray analysis of DRGs from *Islet1* conditional knockout and control embryos have been described (*Dykes et al., 2011*).

## Antibody production and purification
A guinea pig polyclonal antiserum was raised against a MDYKDDDDKLVY peptide that corresponds to the N terminus of Flag-CBFβ encoded by the *Cbfb*$^{Flag}$ allele using a service provided by Covance. The peptide was synthesized and conjugated at its C-terminus to KLH. Exsanguination bleeds were enriched for IgG using Protein A agarose chromatography. A sample of the IgG fraction was further affinity purified using a Flag peptide-conjugated column prepared using the Sulfolink immobilization kit for peptides (Pierce).

## Quantification of epidermal innervation and the intensity of fluorescent or colorimetric signals
For quantifying the extent of epidermal innervation, three randomly selected regions of the epidermis were imaged for each animal. For each image, the epidermal region was defined based on TOPRO3 nuclear stain and selected as the region of interest. Images were then thresholded based

**Table 1.** Primers used for real-time PCR analysis

| Cbfb | F-TCGAGAACGAGGAGTTCTTCAGGA | R-AGGCGTTCTGGAAGCGTGTCT |
|---|---|---|
| Runx1 | F-GCAGGCAACGATGAAAACTACT | R-GCAACTTGTGGCGGATTTGTA |
| Mrgprd | F-TGCTGCTGGAAACACTTCTAGGGA | R-GCTGCTGTCAAGAGTGGAGTTCAT |
| Gfra2 | F-TCGTACAGACCACTTGTGCC | R-ATCAAACCCAATCATGCCAG |
| Ptprt | F-ACCTGCTTCAACACATCACCCAGA | R-TTCATCTTCCTTGGCTGTGTCCCA |
| Myo1a | F-ACAGGTGCTTCAACACAGCCAATC | R-GCCCTTAAACAGTTCACTGGCACA |
| Ret | F-TCAACCTTCTGAAGACAGGCCACA | R-ATGTCAGCAAACACTGGCCTCTTG |
| PGP9.5 | F-CAGACCATCGGAAACTCCTG | R-CACTTGGCTCTATCTTCGGG |
| GAPDH | F-ATGCCTGCTTCACCACCTTCTT | R-ATGTGTCCGTCGTGGATCTGA |

on βIII-Tubulin or GFP immunostaining, and the area fraction (the percentage of pixels above threshold in the region of interest) was calculated using Image J and reported as a measure of epidermal innervation density. Area fraction for each image was considered an individual data point for statistical analysis. For quantifying the intensity of fluorescent images, images except those of cultured neurons were thresholded and regions of interest were defined either on a cell-by-cell basis or as populations of cells. The mean or total intensity of pixels above threshold was measured. For cultured neurons, the total intensity of pixels within each neuron was measured without thresholding. For quantifying in situ hybridization signal intensity, the same procedure was done, except that images were first converted to grayscale.

## Acknowledgements

We thank Dr. Alan Friedman for providing *Runx1^{flox}* mice; Dr. Qiufu Ma and members of the Ginty lab for discussions, technical assistance, and comments on the manuscript; Dori Reimert and Holly Wellington for ES cell work; Lely Quina for preparing the *Islet1* conditional knockout embryos, and Charles Hawkins and the Johns Hopkins Transgenic Facility for blastocyst injections.

## Additional information

### Competing interests

DDG: Reviewing editor, *eLife.* The other authors declare that no competing interests exist.

### Funding

| Funder | Grant reference number | Author |
| --- | --- | --- |
| National Institutes of Health | NS34814 | David D Ginty |
| Whitehall Foundation | 2010-08-61 | Jian Zhong |
| National Eye Institute | 1R01EY022409 | Jian Zhong |
| National Institutes of Health | NS064933 | Eric E Turner |

The funders had no role in study design, data collection and interpretation, or the decision to submit the work for publication.

### Author contributions

SH, Conception and design, Acquisition of data, Analysis and interpretation of data, Drafting or revising the article; KJO'D, EET, JZ, Drafting or revising the article, Contributed unpublished essential data or reagents; DDG, Conception and design, Analysis and interpretation of data, Drafting or revising the article

### Ethics

Animal experimentation: This study was done in accordance with the recommendations in the Guide for the Care and Use of Laboratory Animals of the National Institutes of Health. All of the animals were handled according to approved institutional animal care and use committee (IACUC) protocols of the Johns Hopkins University School of Medicine. The protocols were approved by the Animal Care and Use Committee of the Johns Hopkins University School of Medicine (Protocol Numbers: MO11M10).

## Additional files

### Supplementary files

• Supplementary file 1. Microarray analysis of genes that are differentially expressed in E16.5 DRGs of control and *Runx1 CKO* animals

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
