## [Decision Letter]

Thank you for submitting your work entitled "Extrinsic and intrinsic signals converge on the Runx1/CBFβ transcription factor for nonpeptidergic nociceptor maturation" for peer review at *eLife*. Your submission has been favorably evaluated by Eve Marder (Senior Editor), a Reviewing Editor, and two reviewers. One of the two reviewers, Yves Barde, has agreed to share his identity.

The reviewers have discussed the reviews with one another and the Reviewing editor has drafted this decision to help you prepare a revised submission.

This detailed study by Huang and colleagues addresses how the general class of nonpeptidergic nociceptors becomes defined at a genotypic level during development of sensory neurons within the dorsal root ganglia. Genetically engineered mice, gene profiling, expression assays and in vitro analysis using cultured embryonic neurons are combined to test the temporal impact of NGF/TrkA and Runx1 activity on genes that define the nonpeptidergic lineage. A novel model invoking communication between intrinsic and extrinsic signaling pathways and a key regulatory role for the Runx1/CBFβ complex is well supported.

Its highlight is the finding that NGF/TrkA signalling induces the expression of core binding factor beta (CBFβ) that stabilizes the Runx1 transcription factor. The involvement of the activation of a MAP kinase pathway is convincingly documented not only by pharmacological means but also by demanding experiments involving *Mek* and *Erk* quadruple mutants (2 of them conditional) as well as by a constitutively active *B-Raf* mutant that rescues the CBFβ phenotype in the *TrkA/Bax* mutant. It also appears that CBFβ stabilises Runx1 as impressively documented in Figure 5 (compare u and s).

All in all this is an interesting contribution nicely and convincingly illustrating how growth and transcription factors orchestrate the development of well-defined sub-populations of neurons.

While all were enthusiastic about the study, the manuscript itself needs a thorough go-through for consistency (see below).

Essential revisions:

1) There are some discrepancies between the text and the figures. In particular, some of the figures do not illustrate what is indicated in the text (compare for example Figure 1—figure supplement 1 and the reference to "controls", with unexplained and misleading signal intensities, in the subsection “NGF and Runx1 are similarly required for initiation of nonpeptidergic nociceptor-specific gene expression”). Also, some of the figures are insufficiently labelled and the antigens not identified.

2) The text would also greatly benefit from being better structured and shortened, for example by avoiding numerous repetitions. The summary of the results at the end of the Introduction could be omitted as well as discussion points in the Results section.

3) Leaving out some of the data might also help, in particular those related to the additional genes co-downregulated in the absence of NGF and Runx1 (Figure 1 and Figure 1—figure supplement 1). Functionally nothing is made of *Ptprt, Myo1a* and *Kif21b*.

4) The co-localisation of CBFβ and Runx1 is of course an important point but it is rather poorly illustrated and not quantified (Figure 4 in particular: most of the green and red signals do not overlap). Yet the authors are surely correct, simply because CBFβ is expressed in most neurons and therefore must include the Runx1 sub-population.

5) Previous work by Ugarte et al. may also be quoted as it indicates that the transcription of TRPV1 requires Runx1 and C/EBPβ heterodimers, a related finding that may be of interests to the reader.

6) In some cases (e.g. Figure 2 and Figure 5) a paired, as opposed to an unpaired *t*-test has been used. The reasons for this surprising choice should be explained.

7) Please rephrase or remove the following sentence: “since Ret expression, a readout of general NGF signaling in nonpeptidergic nociceptors…”. This statement is confusing, as it implies that NGF signals through Ret. It can be deleted without changing the point of the sentence.

8) Without supporting data, the statement that proprioceptors are lacking (due to defective Runx3 activity) should be omitted in the Results section (first paragraph of the subsection “CBFβ is required for the development of Runx1-dependent nonpeptidergic populations”). This statement, repeated in the last paragraph of the aforementioned subsection, should also be removed. The effect of CBFβ KO on other *Runx* genes, and the potential loss of proprioceptors could be added to the Discussion.

---

## [Author Response]

Essential revisions:1) There are some discrepancies between the text and the figures. In particular, some of the figures do not illustrate what is indicated in the text (compare for example Figure 1—figure supplement 1 and the reference to "controls", with unexplained and misleading signal intensities, in the subsection “NGF and Runx1 are similarly required for initiation of nonpeptidergic nociceptor-specific gene expression”). Also, some of the figures are insufficiently labelled and the antigens not identified.

Assuming the discrepancy between figures and the text description refers to the unexplained signal for *Myo1a* and *Kif21b* in *NGF^-/-^; Bax^-/-^* and *Runx1 CKO* DRGs in Figure 1—figure supplement 1, the text corresponding to Figure 1 and Figure 1—figure supplement 1 has been modified to make the point that *Myo1a, Kif21b, Gfrα2* are expressed in non-nociceptors and that this is responsible for the residual expression of these genes in *NGF^-/-^; Bax^-/-^* and *Runx1 CKO* DRGs. In addition, images in Figure 1—figure supplement 1 are now shown at higher magnification so that readers can better appreciate the expression deficits in presumptive nonpeptidergic nociceptors of mutant animals. Regarding the rest of the text and figures, in particular Figures 2, 5, 6 and 7, Figure 4—figure supplement 1, Figure 5—figure supplement 1, Figure 6—figure supplement 1, and Figure 7—figure supplement 1, changes have been made as requested.

2) The text would also greatly benefit from being better structured and shortened, for example by avoiding numerous repetitions. The summary of the results at the end of the Introduction could be omitted as well as discussion points in the Results section.

The text, including the summary of the results at the end of the Introduction and discussion points in the Results section, has been carefully examined and edited to streamline the presentation.

*3) Leaving out some of the data might also help, in particular those related to the additional genes co-downregulated in the absence of NGF and Runx1 (Figure 1 and Figure 1—figure supplement 1). Functionally nothing is made of *Ptprt*, *Myo1a* and *Kif21b*.*

The rationale for including expression of additional novel nonpeptidergic nociceptor-specific genes for the phenotypic analysis of *NGF* and *Runx1* mutants is to evaluate the extent to which NGF and Runx1 coregulate nonpeptidergic specific gene expression, a key test of the relationship between NGF and Runx1, which is the foundation of the study. Therefore, we opted to leave these data in the paper. Instead, changes in the text and figure, as mentioned above, were made so that readers can better appreciate the phenotypic similarity of *NGF* and *Runx1* mutants.

4) The co-localisation of CBFβ and Runx1 is of course an important point but it is rather poorly illustrated and not quantified (Figure 4 in particular: most of the green and red signals do not overlap). Yet the authors are surely correct, simply because CBFβ is expressed in most neurons and therefore must include the Runx1 sub-population.

The reviewers are correct that the highest green and red signals generally do not overlap. In fact, a co-localization analysis (data not shown) suggests an anti-correlated relationship between the two, which along with the observation that CBFβ expression was found to be higher in *Runx1* mutant DRGs relative to control (data not shown), suggests negative cross-regulation of CBFβ expression by Runx1. This interesting observation does not negate the fact that green signals, above background as seen in wildtype DRGs (Figure 4), are always found where red signals are, leading to the conclusion of Runx1 and CBFβ colocalization in DRG neurons.

5) Previous work by Ugarte et al. may also be quoted as it indicates that the transcription of TRPV1 requires Runx1 and C/EBPβ heterodimers, a related finding that may be of interests to the reader.

While we agree that the aforementioned study demonstrated *Trpv1* as a direct transcriptional target of Runx1 and C/EBPβ heterodimers (which is not to be mistaken for CBFβ, a subunit of the functional Runx1 transcription factor complex) in dorsal root ganglion neurons, this finding has little bearing on our study, which has identified the Runx1/ CBFβ complex as a point of convergence between extrinsic and intrinsic factors that control nonpeptidergic nociceptor maturation.

*6) In some cases (e.g. Figure 2 and Figure 5) a paired, as opposed to an unpaired *t*-test has been used. The reasons for this surprising choice should be explained.*

We have repeated the relevant analyses using unpaired t tests, and the figures and legends have been updated accordingly. The statistical significance and conclusions hold true regardless of the type of t test used.

7) Please rephrase or remove the following sentence: “since Ret expression, a readout of general NGF signaling in nonpeptidergic nociceptors…”. This statement is confusing, as it implies that NGF signals through Ret. It can be deleted without changing the point of the sentence.

The text was changed to avoid confusion.

*8) Without supporting data, the statement that proprioceptors are lacking (due to defective Runx3 activity) should be omitted in the Results section (first paragraph of the subsection “CBFβ is required for the development of Runx1-dependent nonpeptidergic populations”). This statement, repeated in the last paragraph of the aforementioned subsection, should also be removed. The effect of CBFβ KO on other *Runx* genes, and the potential loss of proprioceptors could be added to the Discussion.*

The text was changed according to the reviewers' recommendation.